

SciPost Phys. Lect. Notes 91 (2025)

# Two lectures on Yang-Lee edge singularity and analytic structure of QCD equation of state

**Vladimir V. Skokov**

Department of Physics, North Carolina State University, Raleigh, NC 27695, USA

vskokov@ncsu.edu

## Abstract

These lecture notes, prepared for the 2024 XQCD PhD, provide an introduction to the analytic structure of an equation of state near a second-order phase transition and its most prominent landmark: the Yang-Lee edge singularity. In addition to discussing general properties, the notes review recent theoretical progress in locating the QCD critical point by tracking the trajectory of the Yang-Lee edge singularity.

| | |
|---|---|
| Received | 2024-11-12 |
| Accepted | 2025-02-25 |
| Published | 2025-03-07 |

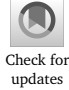

# 1   Introduction and motivation

In these lecture notes, I discuss the analytic structure of the QCD phase diagram focusing on potential phase transitions and critical points. When I refer to the analytic structure, I am simply addressing the existence and positioning of poles, branch points, and branch cuts of pressure (or, more generally, the partition function) in the complex plane of thermodynamic variables (such as the complex baryon chemical potential in the context of QCD). The groundbreaking work of Lee and Yang highlighted a profound link between the analytical structure of the equation of state and the phase structure [1,2]. Their primary focus was on finite volume systems. In these notes, I will take a different approach, predominantly considering the thermodynamic limit with only sporadic mentions of finite volume systems.

In my experience, a fraction of the audience loses interest whenever I talk about complex values of thermodynamic variables. There is a widespread perception that going to the complex plane is a purely academic problem with no real-world application to physical systems. That is why it is crucial to address the significance of the analytic structure of the equation of state right from the start. Firstly, from a theoretical perspective, we aim to learn about QCD in all possible regimes. As an extreme example, it is known that some deformations of QCD might allow for an analytic treatment of confinement (see e.g. Ref. [3]). Going to complex values of baryon chemical potential alone is not sufficient to carry out this program, but it is complementary as it helps to establish if it is possible to remove the deformation without crossing phase transitions and critical points. Secondly, there is a pragmatic reason, as far as QCD is concerned. The main tool to study the QCD phase diagram non-perturbatively and from the first principles is lattice QCD.[1] At non-zero *real* values of the chemical potential, the lattice approach suffers from the so-called sign problem which prohibits efficient importance sampling and renders the approach unpractical. This is the main reason why our knowledge of QCD phase diagram at finite chemical potential is limited. Despite this problem, performing lattice QCD calculations at zero or purely imaginary chemical potential can still yield quite a bit of information. Let's discuss the former. The idea is to compute the Maclaurin series expansion of pressure

$$p(\mu, T) \approx p(0, T) + \frac{\partial p(0, T)}{\partial \mu}\mu + \frac{1}{2}\frac{\partial^2 p(0, T)}{\partial \mu^2}\mu^2 + \frac{1}{3!}\frac{\partial^3 p(0, T)}{\partial \mu^3}\mu^3 + \frac{1}{4!}\frac{\partial^4 p(0, T)}{\partial \mu^4}\mu^4 + \dots \quad (1)$$

Note that all derivatives here are evaluated at zero baryon chemical potential and therefore, are computable on the lattice! The above expansion can be simplified by taking into account the quark-anti-quark symmetry of the QCD partition function $p(\mu, T) = p(-\mu, T)$; which sets the odd-order derivatives to zero at zero chemical potential, such that the expansion only involves even powers of $\mu$:

$$p(\mu, T) \approx p(0, T) + \frac{1}{2}\frac{\partial^2 p(0, T)}{\partial \mu^2}\mu^2 + \frac{1}{4!}\frac{\partial^4 p(0, T)}{\partial \mu^4}\mu^4 + \dots \quad (2)$$

---

[1]Note that functional methods have become very prominent recently, see, e.g., Refs. [4,5].

This expansion may potentially allow us to study the function $p(\mu, T)$ at any value of $\mu$. To illustrate this, let's consider two simplified models mimicking some of the properties of QCD in high- and low-temperature limits. At low $T$, the thermodynamics of QCD is well described by the Hadron Resonance Gas model (a collection of non-interacting hadrons). We are only interested in baryonic degrees of freedom here, as only they depend on the chemical potential. Due to the substantial mass of baryons, we also perform degeneracy expansion (see e.g. Appendix A1.4 of Ref [6]) and keep only the leading order for illustrative purpose.[2] For the pressure (neglecting chemical potential independent terms), we obtain

$$p^{\mathrm{HRG}}(\mu_B, T) = C \cosh(\hat{\mu}_B),$$

where $\hat{\mu}_B = \mu_B/T$ and $C$ is a constant whose form is irrelevant to our discussion. The series expansion of $\cosh(x) = \sum_{i=0}^{\infty} \frac{x^{2i}}{(2i)!}$ is known to an arbitrary order, with the series converging for any value of $x$. This can be tested by evaluating the radius of convergence

$$R = \frac{1}{\lim_{n\to\infty}(c_n)^{1/n}} = \frac{1}{\lim_{n\to\infty}\left(\frac{1}{(2n)!}\right)^{1/n}} = \infty. \tag{3}$$

Practically, this means that if one wants to compute a value of the function $p^{\mathrm{HRG}}(\mu_B, T)$ at a some $\mu_B = \mu_B^*$ with precision $\delta$, there exists such an $n$ for which the truncated Maclaurin series is an appropriate approximation, that is $|p^{\mathrm{HRG}}(\mu_B^*, T) - C\sum_{i=0}^{n} \frac{\hat{\mu}_B^{2i}}{(2i)!}| < \delta$. It is worth mentioning that the function $\cosh(x)$ is analytic in the complex plane of $x$, thus it does not have singularities at finite (complex) values of $x$.

As the reader suspects, the situation is not that simple in QCD. Our next example of free quarks illustrates this. Consider the corresponding pressure (see Chapter 2 of Ref [6])

$$p^{\mathrm{SB}}(\mu, T) = \frac{2N_c N_f}{3} \int \frac{d^3p}{(2\pi)^3} \frac{p^2}{\omega} \frac{1}{e^{\hat{\omega}-\hat{\mu}}+1} + (\mu \to -\mu),$$

where $\hat{\omega} = \sqrt{p^2 + m^2}/T$. In principle, one can expand this function in $\mu$ around zero and analyze the convergence. However, for simplicity's sake, let us consider the integrand only and analyze the corresponding series expansion. We get

$$\frac{1}{e^{\hat{\omega}-\hat{\mu}}+1} + (\hat{\mu} \to -\hat{\mu}) = \frac{2}{e^{\hat{\omega}}+1} + 2\sinh^4\left(\frac{\hat{\omega}}{2}\right)\operatorname{csch}^3(\hat{\omega})\hat{\mu}^2 + O\left(\hat{\mu}^4\right). \tag{4}$$

Higher orders can be readily computed, but their analytic form becomes rather cumbersome. I will instead plot this function and its truncated Maclaurin series approximations in Fig. 1. In this figure, we see that beyond a certain value of $\hat{\mu}$, going to higher orders in expansion does not improve the description of this function. An analysis of the convergence radius, see Eq. (3), gives a finite number $R \approx 3.3$. Now, let's analyze this function in the complex plane. It is sufficient to consider only one term (Fermi distribution) $f(\hat{\mu}) = \frac{1}{e^{\hat{\omega}-\hat{\mu}}+1}$. The denominator of this function may go to zero at some value of $\hat{\mu}$, resulting in a pole of $f(\mu)$. Simple algebra yields

$$\hat{\mu}^* = \hat{\omega} \pm i\pi n,$$

where $n$ is a positive odd integer. The origin of $n$ is due to the periodicity of our function: $f(\hat{\mu}) = f(\hat{\mu} + 2\pi i)$, which can be proven easily. Specifically, the singularities closest to the origin are located at $\hat{\mu}^* = \hat{\omega} \pm i\pi$ which, for $\hat{\omega} = 1$ are at a distance $\sqrt{1+\pi^2} \approx 3.3$ away from

---

[2]Note that this approximation is violated at high values of the baryon chemical potential. When all order of the degeneracy expansion are accounted for, the analytic structure is similar to the that of free quarks with $\mu \to \mu_B$.

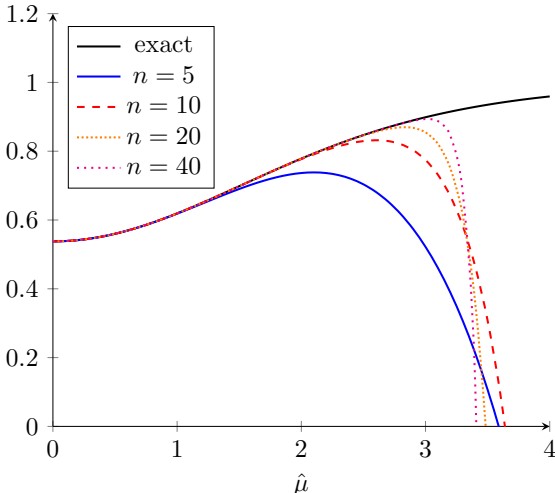

Figure 1: $n$-th order truncated Maclaurin series approximations vs the function itself for $\hat\omega = 1$.

the origin. This is a simple illustration of an important statement: *the radius of convergence of a series represents the distance in the complex plane from the expansion point to the nearest singularity of the function expanded.*[3]

Before we move on, I want to draw your attention to an interesting fact about the Fermi distribution function when evaluated on the line $\hat\mu = \hat\mu_r + i\pi$:

$$f(\hat\mu) = \frac{1}{e^{\hat\omega - \hat\mu_r + i\pi} + 1} = \frac{1}{1 - e^{\hat\omega - \hat\mu_r}} = -f_B(\hat\mu_r), \tag{5}$$

where $f_B$ is the Bose distribution and $\hat\mu_r$ is purely real. This equality demonstrates that the Fermi distribution morphs into a negative Bose distribution which can develop singularity due to Bose-Einstein condensation for $\hat\mu_r = \hat m$.

To summarize, we demonstrated that in the case of $p^{\mathrm{SB}}(\mu, T)$[4] what we can learn from series expansions is limited[5] with singularity in the complex plane defining the horizon of our knowledge. Let me stress again: this is the main reason we want to know the analytic structure of QCD.

## 2 Analytic structure near a critical point: Mean-field approximation

The key question is whether phase transitions imprint the analytic structure of pressure or, in general, the partition function. To explore this, we consider Landau's mean-field theory of phase transitions. As far as the long-range order is concerned, the mean-field approximation becomes exact above the upper critical dimension, as I explain in the next section. For the physical case of three spatial dimensions, the mean-field theory can only roughly approximate a system near a second-order phase transition.

---

[3]The generic proof is a byproduct of one of the most important theorems of complex analysis "a holomorphic function is analytic and vice versa".

[4]We only analyzed the integrand of this function. The integral will evolve a sum over a continuum of poles. This sum leads to a branch point singularity $\hat\mu^* = \hat m \pm i\pi n$ and associated cuts.

[5]In Sec. 5.2.2, I will return to this and discuss how the knowledge of the analytic structure can help us learn more about the function.

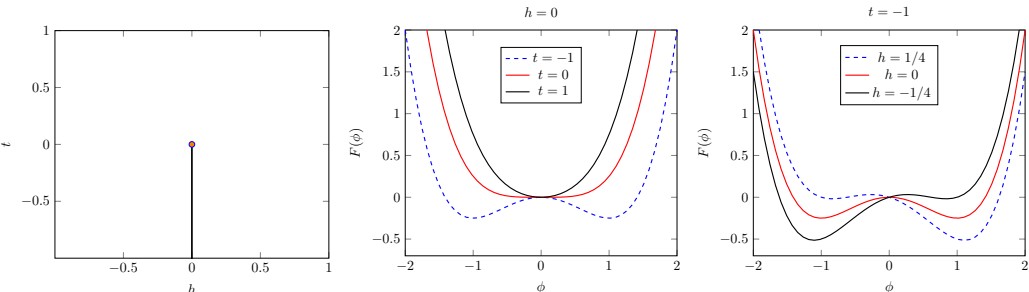

Figure 2: Phase diagram of the mean-field Landau model and two types of phase transitions. The middle panel illustrates the second-order phase transition at zero external magnetic field $h = 0$. The right panel shows the first-order phase transition for a fixed negative value of $t = -1$.

## 2.1 Landau's mean-field theory

Landau's mean-field model is defined by the following free energy:

$$F = \frac{t\phi^2}{2} + \frac{\lambda\phi^4}{4} - h\phi \,. \tag{6}$$

To keep the theory stable, here we assume that $\lambda$ is a positive constant. Therefore an appropriate rescaling of the field $\phi$ and parameters $t$ and $h$ is equivalent to setting $\lambda = 1$. $t$ is the so-called reduced temperature $t = (T - T_c)/T_c$ and $h$ is the external field. At zero $h$ the Landau free energy has $Z(2)$ symmetry ($F(\phi) = F(-\phi)$). In contrast to $t$, nonzero values of $h$ break this symmetry explicitly. Analyzing the potential for different values of the parameters allows for easy reconstruction of the model's phase diagram. I present an illustration for two different transition scenarios in Fig. 2. There is a third possible transition scenario, which in the heavy-ion collision community is known by the name of "crossover".[6] It denotes a smooth transition at a fixed positive value of $t$ as $h$ changes sign.

The expectation value of the order parameter $\phi$ can be found by minimizing $F$. Before attacking the most generic case, let's first consider two special cases: zero $h$ and zero $t$.

- At zero $h$, the minimization of the free energy leads to the following values of the order parameter:

$$\phi = \begin{cases} 0, & \text{for } t \geq 0, \\ \pm(-t)^{1/2}, & \text{for } t < 0. \end{cases} \tag{7}$$

  This shows that (in infinite volume) the symmetry is spontaneously broken (the system has either a positive or negative value of the order parameter) when $t < 0$ and the order parameter develops a non-zero expectation value. Generically we have $\phi = (-t)^{\beta}$, where $\beta$ is a universal critical exponent (in Landau theory, $\beta = 1/2$).

- At zero $t$, minimization of $F$ leads to

$$\phi = h^{1/3} \,. \tag{8}$$

  At critical temperature $t = 0$, the dependence of the order parameter on the external field defines the critical exponent $\delta$: $\phi = h^{1/\delta}$.

---

[6]It is not related to the crossover phenomenon (see Ref. [7] for details) of critical statics.

## 2.2 Singularities of Landau mean-field theory

We are ready to analyze the equation of motion in the general case:

$$t\phi + \phi^3 - h = 0. \tag{9}$$

Here we specifically want to investigate if the order parameter has a singularity in the complex plane of $t$ or $h$. For the sake of the argument, let's fix $t > 0$ and consider $\phi$ as a function of $h$. The analysis can also be performed for complex values of $t$, but it will not provide any additional insights, as it will be clear below.

In Eq. (8), we saw that at the singularity/critical point the order parameter behaves as $\phi \propto h^{1/\delta}$, where $|\delta| > 1$. Generically, if a singularity is not at a zero magnetic field, we have $\phi \propto (h - h_c)^{1/\delta}$. Let's consider this as an Anzatz. Then at $h = h_c$, we get

$$\frac{d\phi}{dh} = \infty,$$

or equivalently

$$\frac{dh}{d\phi} = 0.$$

To reformulate it more conveniently, let's consider the equation of motion in its implicit form and differentiate with respect to h, yielding

$$\frac{d}{dh}\frac{\partial F}{\partial \phi} = \frac{\partial^2 F}{\partial \phi^2}\frac{d\phi}{dh} + \frac{\partial^2 F}{\partial \phi \partial h} = 0. \tag{10}$$

We have $\frac{\partial^2 F}{\partial \phi \partial h} = -1$ and thus

$$\frac{\partial^2 F}{\partial \phi^2}\frac{d\phi}{dh} = 1, \tag{11}$$

inferring that the location of the singularity can be found by requiring that the mass of the order parameter vanishes:

$$\frac{\partial^2 F}{\partial \phi^2} = t + 3\phi^2 = 0. \tag{12}$$

The easiest way to proceed is to multiply this equation by $\phi/3$ and subtract the result from the equation of motion to get

$$\phi_c = \frac{3}{2}\frac{h_c}{t}. \tag{13}$$

Substituting this into Eq. (12) leads to

$$h_c = \pm i\frac{2}{3\sqrt{3}}t^{3/2}. \tag{14}$$

It is worth pausing here to contemplate the consequences of this result. First, we found that, indeed, for positive $t$, there are singularities in the complex $h$ plane; the singularities are located at purely imaginary values of $h$. Second, we found that a simple combination $h/t^{3/2}$ or equivalently $z = t/h^{2/3}$ is a pure number ($z_c = \frac{3}{2^{2/3}}e^{\pm i\pi/3}$) independent of $t$ at the singularity. This hints at a certain symmetry in the equation of motion, which we will now identify.

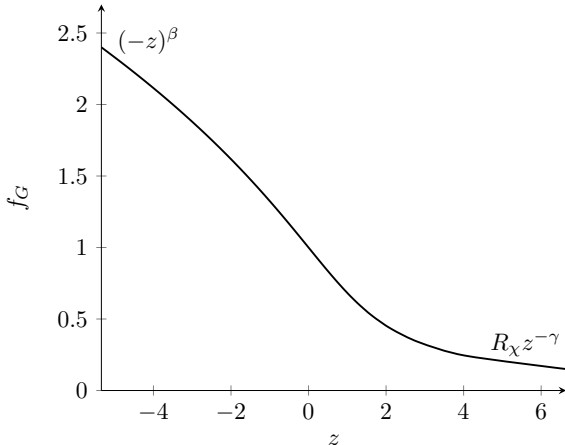

Figure 3: The magnetic equation of state in the mean-field approximation.

## 2.3 Magnetic equation of state

Let's revise the equation of motion by using the following ansatz for the solution $\phi = h^{1/\delta} f_G(t, h)$ with $\delta = 3$:

$$t h^{1/3} f_G + h f_G^3 - h = 0,$$

$$\frac{t}{h^{2/3}} f_G + f_G^3 - 1 = 0.$$

This demonstrates that $f_G$ is a function of a combination $z = t/h^{2/3}$ known as a *scaling variable*. The equation defining $f_G$ is called the magnetic equation of state. For a general universality class and beyond mean-field approximation, one can show (see e.g. [7]) that the scaling variable is given by $z = t/h^{\frac{1}{\beta\delta}}$, where $\beta$ and $\delta$ are the critical exponents of the corresponding universality class. Moreover, in contrast to the mean-field results, the magnetic equations of state are not known analytically (with rare exceptions, e.g., infinite $N$ limit).

Figure 3 shows the magnetic equation of state (note that you do not have to solve for $f_G$ to plot it; instead, plot the inverse function $z = 1/f_G - f_G^2$). The conventional way to properly define the magnetic equation of state is to rescale $t$ and $h$[7] in such a way as to satisfy

$$f_G(z = 0) = 1, \tag{15}$$

$$f_G(z \to -\infty) = (-z)^\beta. \tag{16}$$

Our mean-field result already satisfies these requirements (if we considered $\lambda \neq 1$, it would not be the case). The asymptotic behavior of $f_G$ at large positive value of the argument is governed by $R_\chi z^{-\gamma}$, where $R_\chi$ is a universal amplitude ratio (in mean-field, $R_\chi = 1$) and $\gamma$[8] is a universal susceptibility critical exponent (in the mean-field, $\gamma = 1$).

As we established, function $f_G$ has two complex conjugate singularities (and associated cuts) in the complex $z$ plane. They are illustrated in Fig. 4. These are the Yang-Lee edge singularities [10]. The first part of the name is due to an intimate connection between these singularities and Lee-Yang zeroes[9] introduced in Refs. [1] and [2]. The second part of the name

---

[7]This is achieved by introducing the so-called metric factors, see e.g., Ref. [8] for details.

[8]Note that there are only two independent critical exponents for the critical point. There is an exact (scaling) relation between $\beta, \delta$ and $\gamma$: $\gamma = \beta(\delta - 1)$. The origin of this and other scaling relations can be explained by the renormalization group; see Ref. [7,9] for a detailed introduction.

[9]Note that the order of the names is switched. Together, Lee and Yang published two papers on the subject, and the papers had different author orderings.

– "edge" refers to the fact that in the finite volume, the cut splits into a countable number of poles of $f_G$ with the first pole approximating the location of singularity. Therefore, the singularity defines the edge where the locus of poles originates (or terminates, if you prefer).

The magnetic equation of state, as defined above, possesses the property of universality. That is, it is independent of short-range interactions and is fully determined by the global symmetries and the number of spatial dimensions. It follows that the locations of the singularities of the magnetic equation of state are likewise universal.

It is instructive to return to Eq. (14), and plot a three-dimensional phase diagram by considering real and imaginary values of $h$, see Fig. 5. For now, let's focus on the mean-field result (left panel) and the "crossover" region: $t > 0$. In the three-dimensional diagram, the Yang-Lee edge singularities form lines (each point of the lines is a Yang-Lee edge). These lines cross at the critical point. In that sense, the critical point has higher co-dimension than the locus of Yang-Lee edge singularities. Most importantly, Fig. 5 demonstrates that *Yang-Lee edge singularity is continuously connected to the critical point*. Thus, tracing the singularity allows one to locate the corresponding critical point.

## 2.4 Investigating the Yang-Lee edge singularity

To what type of singularity does a Yang-Lee edge belong? In Figure 4 we already saw that this is a branch point. Let's demonstrate this explicitly. We start from the equation for the expectation value of the order parameter Eq. (9), and we expand it near the Yang-Lee edge:

$$\phi = \phi_c + \tilde{\phi}\,, \tag{17}$$

where $\phi_c$ we found previously in Eq. (13) and we consider $\tilde{\phi}/\phi_c \ll 1$. We obtain (after collecting terms)

$$\underbrace{t\phi_c + \phi_c^3}_{=h_c} + \underbrace{(t + 3\phi_c^2)}_{=0}\tilde{\phi} + 3\phi_c\tilde{\phi}^2 + \tilde{\phi}^3 - h = 0\,. \tag{18}$$

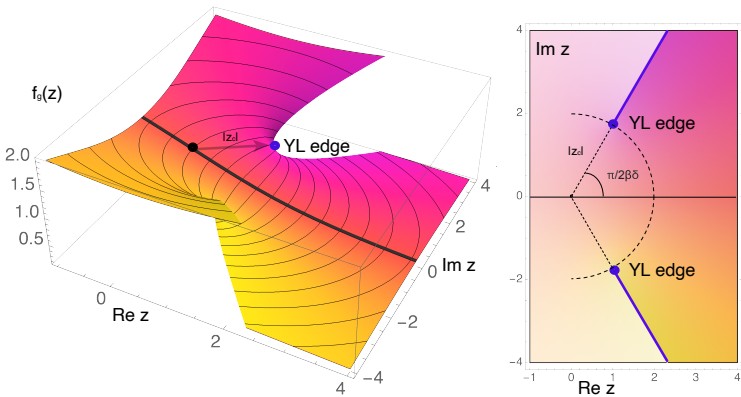

Figure 4: The real part of $f_G(z)$ as a function of complex $z$. The black solid line in the figure on the left panel is the magnetic equation of state for real values of $z$; see also Fig. 3.

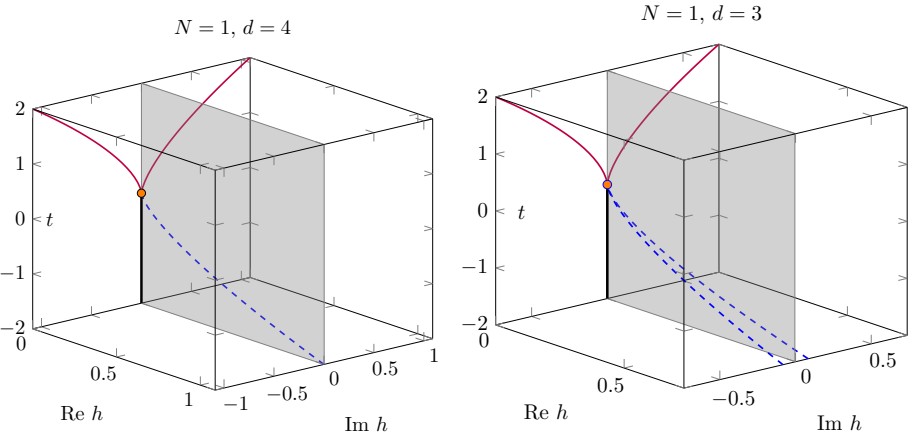

Figure 5: Phase diagram of Ising model in mean-field approximation (or $d = 4$) and with fluctuations taking into accounts.

Where the first term gives $h_c$ due to equations of motion, while the second term is zero due to zero mass term at Yang-Lee edge, see Eq. (12). Finally, we can safely neglect $\tilde{\phi}^3$ term. We thus end up with a simple result:

$$\tilde{\phi} = \left(\frac{h - h_c}{3\phi_c}\right)^{1/2}, \tag{19}$$

$$\phi = \phi_c + \left(\frac{h - h_c}{3\phi_c}\right)^{1/2}. \tag{20}$$

This shows that, in the mean-field approximation, the Yang-Lee edge is an algebraic branch point. The behavior of the order parameter near the singularity defines the edge critical exponent $\sigma$. Thus, as we just showed, in the mean-field, $\sigma_{\mathrm{MF}} = 1/2$. The edge singularities can be seen as critical points themselves. As shown in Fig. 5, by changing only one parameter, $h$, we can adjust the system to these critical points. In contrast, the conventional critical point requires tuning two parameters, $t$ and $h$ (relevant variables). One would need to adjust four or more parameters to reach a multi-critical point, such as a tricritical point [11]. For this reason, M. Fisher dubbed Yang-Lee edge singularities *protocritical* points [12]. The number of relevant variables also determines the number of independent critical exponents in the leading order scaling relations.[10] At the conventional critical point, two relevant perturbations result in two independent critical exponents. On the other hand, at the Yang-Lee edge, we only have one independent critical exponent, $\sigma$.

Note that from the definition of $\sigma$, one sees its relation to $\delta_{\mathrm{YLE}} = 1/\sigma_{\mathrm{YLE}}$. It is important to stress that $\delta$ of the underlying universality class does *not* have to be (and often not) equal to $\delta_{\mathrm{YLE}}$. As the reader might have correctly guessed, the Yang-Lee edge belongs to a different universality class from the conventional critical point of the underlying theory. We will discuss this in the next section. Before we proceed, let's summarize our findings.

## 2.5 A brief summary

- Yang-Lee edge singularity is continuously connected to the critical point. Thus, by tracing the trajectory of the singularity as a function of temperature, one can establish the existence and the location of the corresponding critical point.

---

[10]In these notes, we only consider the minimal set of critical exponents that are required for the narrative. To learn more about a complete set of them and the algebraic relations connecting them, see Ref. [7].

- Yang-Lee edge is an algebraic branch point singularity.

- When expressed in terms of the scaling variable, the location of the singularity is universal.

- This singularity limits (when it is the closest to the expansion point) the radius of convergence of a series.

There are some questions one might have concerning the list above.

- Would the existence of Yang-Lee edge singularity infer the existence of the corresponding critical point? The answer to this question is a qualifying no. Let me explain by considering a one-dimensional Ising model as an example:

$$Z = \sum_{\{\sigma\}} e^{-\frac{H(\sigma)}{T}} = \sum_{\sigma} e^{\hat{J} \sum_{i=1}^{L-1} \sigma_i \sigma_{i+1} + \hat{h} \sigma_i} \,. \tag{21}$$

The free energy of the model is known exactly [13]:

$$f = -\lim_{L \to \infty} \frac{T}{L} \ln Z = -T \ln \left( e^{\hat{J}} \cosh(\hat{h}) + \sqrt{e^{2\hat{J}} \sinh^2(\hat{h}) + e^{-2\hat{J}}} \right), \tag{22}$$

where the shorthand notation $\hat{x} = x/T$ was used. We expect to have an algebraic branch point, and thus it is natural to check when the argument of the square root crosses zero:

$$e^{2\hat{J}} \sinh^2(\hat{h}_c) = -e^{-2\hat{J}} \,, \tag{23}$$

or

$$h_c = \pm i T \arcsin e^{-2\hat{J}} \,. \tag{24}$$

As expected, we obtained two complex conjugate singularities. They coincide/cross when the imaginary part goes to zero, which is only possible for a fixed value of $J$ when $\hat{J}$ goes to infinity and thus $T \to 0$. However, a critical point at $T = 0$ is a quantum one; the classical Ising model is not appropriate for describing a system in this limit.

We thus see that the existence of the Yang-Lee edge singularities does not indicate the existence of the *finite* temperature critical point. However, the opposite statement is true: every finite temperature critical point has the associated pair of Yang-Lee edges [10].

As a by-product of our calculation, expanding the free energy around Yang-Lee edge singularity, we obtain:

$$f = T \ln \left( e^{\hat{J}} \sqrt{1 - e^{-4\hat{J}}} \right) + T \frac{(1+i) e^{-\hat{J}}}{(1 - e^{-4\hat{J}})^{1/4}} (\hat{h} - \hat{h}_c)^{1/2} + \mathcal{O}((h - h_c)^{3/2}), \tag{25}$$

or simply

$$f - f_c \propto (\hat{h} - \hat{h}_c)^{1/2} + \mathcal{O}((h - h_c)^{3/2}). \tag{26}$$

Therefore, the order parameter (the magnetization) is given by

$$\phi - \phi_c \propto (\hat{h} - \hat{h}_c)^{-1/2} + \mathcal{O}((h - h_c)^{1/2}). \tag{27}$$

We thus established that in one spatial dimension, the edge critical exponent is negative $\sigma_{d=1} = -1/2$.

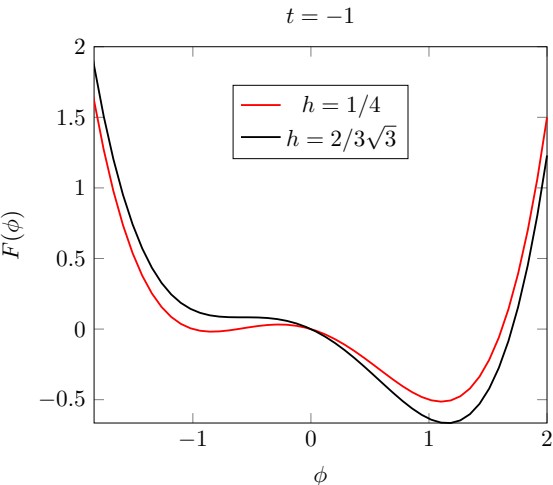

Figure 6: The Landau potential close and at the spinodal point.

- What is the fate of the Yang-Lee edge singularities in the low temperature phase $t < 0$?

  In this regime, Yang-Lee edge singularities are known as spinodals, see also Refs. [14,15]. At a given $t < 0$, the spinodal point is defined by the limiting value of $h$ at which the system loses its local stability against phase separation concerning small fluctuations. Figure 6 illustrates the system close and at the spinodal point. Note that spinodal lines lie on the real $h$ axis only in the mean-field approximation. Recall that we have

$$h_c = \pm i |z_c|^{-\beta\delta} t^{\beta\delta}. \tag{28}$$

  For negative $t$ and non-integer or non-half-integer $\beta\delta$, $it^{\beta\delta} = i|t|^{\beta\delta} e^{\pm i\pi\beta\delta}$ is a complex number. However, for half-integer $\beta\delta$ (as we have in mean-field), this combination becomes purely real; see Fig. 5 for illustration. Thus, as far as the *real* values of the thermodynamic parameters are concerned, a spinodal is a mean-field object and does not exist in "real" systems in thermodynamic equilibrium.

## 3 Beyond mean-field approximation

### 3.1 Upper critical dimension

The previous section discussed most of the systems in the mean-field approximation framework (except for the one-dimensional Ising model). However, going beyond it and accounting for fluctuations is not a simple task in general but is often essential to describe a system near a critical point. The role of fluctuations depends sensitively on the number of spatial dimensions. I will motivate below that there exists such a $d_c$ - the dimension where fluctuations become important - above which $d \geq d_c$ the mean-field approximation is sufficient to describe long-range order. $d_c$ is usually referred to as the upper critical dimension. Our first goal is to establish the upper critical dimension in the vicinity of the critical point and Yang-Lee edge singularity.

Once we account for fluctuations, most of the universal properties are not computable analytically. Moreover, in contrast to mean-field approximation when the number of spatial dimensions, $d$ and the number of the field components, $N$ never explicitly entered into any of our calculations, the universal properties (e.g., critical exponents, the position of the singular-

ity $z_c$) now depends on (and often *only* on) $d$ and $N$.[11]

Let's consider our theory near the Yang-Lee edge to motivate further discussion. We start with the Ginzburg-Landau Lagrangian

$$L = (\nabla \phi)^2 + \frac{1}{2} t \phi^2 + \frac{\lambda}{4} \phi^4 - h \phi \,. \tag{29}$$

This Lagrangian describes the system in the vicinity of the critical point. We can now shift the field variable $\phi = \phi_c + \varphi$. Performing the expansion, we get

$$L = (\nabla \varphi)^2 + \frac{1}{2}(t + 3\lambda \phi_c^2)\varphi^2 - \lambda \phi_c \varphi^3 - (h + t \phi_c + \lambda \phi_c^3)\varphi \,. \tag{30}$$

Here, I neglected the $\varphi^4$ as it is irrelevant given the presence of the cubic interaction term. Now we can fix $\phi_c$ by requiring the absence of the mass term (i.e., tune the system to the vicinity of the Yang-Lee edge; note that for $t > 0$, zero mass leads to purely imaginary $\phi_c$). We get

$$L \approx (\nabla \varphi)^2 - \lambda \phi_c \varphi^3 - H \varphi \,, \tag{31}$$

with $H = h + t \phi_c + \lambda \phi_c^3$. I want to highlight a few interesting observations about the obtained Lagrangian. First, the coupling constant $-\lambda \phi_c$ is purely imaginary due to the presence of the factors $\phi_c$. Second, and most important, we obtained a $\varphi^3$ theory (not $\phi^4$). To understand why it matters, we must take a short detour into the QFT renormalization theory.

In QFT, theories are classified into non-renormalizable, renormalizable, and super-renormalizable. The theory is renormalizable if the coupling constant is dimensionless [16]. In QFT, the renormalization technique is used to treat UV behavior. In contrast, the theory of critical phenomena is concerned with infrared behavior. Below, we demonstrate how these regimes are connected.

Let's determine the canonical dimension of the coupling constant in a theory with interaction $\lambda_r \phi^r$ in $d$ dimension. The field $\phi$ has dimension of $\Lambda^{\frac{d-2}{2}}$, where $\Lambda$ denotes the units of inverse length (or momentum). This can be easily determined by considering the kinetic term and remembering that the action is dimensionless (in natural units $c = \hbar = 1$). For the coupling constant we thus have $[\lambda_r]\Lambda^{\frac{d-2}{2}r}\Lambda^{-d} = 1$ or $[\lambda_r] = \Lambda^{d+r-\frac{1}{2}rd}$. Therefore, the coupling constant of $\lambda \phi^4$ theory is dimensionless in $d = 4$, while the one of $\lambda \phi^3$ theory in $d = 6$. The theories are also renormalizable in the same number of dimensions.

For $\phi^4$ interaction, the theory is super-renormalizable for $d < 4$ from the QFT perspective, while from the standpoint of critical phenomena, this is where the theory requires the most complex calculations. Vice versa, for $d > 4$, the theory is non-renormalizable in QFT, but in statistical physics, its description is the simplest: Gaussian approximation/mean-field.

To motivate why this is the case, let's consider a simple graph in $\phi^3$ theory:[12]

$$I(k) = \quad\quad = \int d^d q \frac{1}{(q^2 + t)((k-q)^2 + t)} \,. \tag{32}$$

Field theoretically, $t$ is the mass squared. We want to analyze the behavior of this integral in the infrared by rescaling the external momentum and the mass squared $k \to \alpha k$ and $t \to \alpha^2 t$ with $\alpha \to 0$. For $d > 4$, $I(k)$ is UV divergent, and although we are not interested in its UV

---

[11]The edge critical exponent $\sigma$ is independent of $N$ of the underlying universality class. This is due to explicit symmetry breaking at non-zero values of $h$, which singles out one critical mode. $\sigma$ does depend on the number of dimensions as we already showed that $\sigma_{d=1} = -1/2$ and $\sigma_{d\geq 6} = 1/2$.

[12]This part is based on Ref. [7].

structure, it presents a technical difficulty for analyzing the IR. To isolate IR, we will perform the following trick. First, we insert

$$\frac{1}{d}\sum_{i=1}^{d}\frac{\partial q_i}{\partial q_i}=1 \tag{33}$$

into the integrand. Second, we perform the integration by parts. This will give two distinct contributions

$$I(k)=\frac{1}{d}\int_S \frac{\vec{q}\cdot d\vec{s}}{(q^2+t)((k-q)^2+t)}+\frac{1}{d}\int d^d q \frac{2}{(q^2+t)((k-q)^2+t)}\left[\frac{q^2}{q^2+t}-\frac{\vec{q}\cdot(\vec{k}-\vec{q})}{(k-q)^2+t}\right]. \tag{34}$$

The first term is a surface contribution evaluated in the UV, $|q|\sim\Lambda$ (UV cutoff). This contribution is analytic for vanishing $k$ and $t$. We can safely drop it. The remaining contribution can be rewritten as

$$I(k)=\frac{2}{d}\left[2I(k)-\int\frac{1}{(q^2+t)((k-q)^2+t)}\left(\frac{t}{q^2+t}+\frac{\vec{k}\cdot(\vec{k}-\vec{q})+t}{(k-q)^2+t}\right)\right], \tag{35}$$

or

$$I(k)=\frac{2}{4-d}\left[\int\frac{1}{(q^2+t)((k-q)^2+t)}\left(\frac{t}{q^2+t}+\frac{\vec{k}\cdot(\vec{k}-\vec{q})+t}{(k-q)^2+t}\right)\right]. \tag{36}$$

This integral is now convergent for $d < 5$. This procedure can be continued until one ends up with UV convergent integrals in which one can set $\Lambda\to\infty$, thus removing an extra scale. The resulting expression will be a homogeneous function of $\alpha$ (since the only present scales are $k$ and $t$). The degree of the homogeneity of the graph is defined by the graph's dimension, $\mathfrak{d}$. Its calculation is identical to the one in QFT; thus I will only briefly outline it.

Again, we will consider $\lambda_r\phi^r$ interaction in $d$-dimensions and focus on computing the dimension of $n$-th order vertex with $E$ external legs. The graph behaves as $\alpha^{\mathfrak{d}}$ with $\mathfrak{d}=Ld-2I$, where $L$ is the number of loops (loop integrations) and $I$ is the number of the internal lines (dimension of each propagator is $-2$). We can express the number of loops $L$ as the number of internal lines minus the number of momentum conservation conditions. Each interaction brings a momentum-conserving conserving delta-function, but one of them should not be included as it is responsible for the overall momentum conservation, thus $L=I-(n-1)$. The number of the internal lines is given by the number of all lines from the interaction vertices ($nr$) minus the number of the connected to the external legs. However, it is important to remember that two lines form a propagator, thus $I=\frac{nr-E}{2}$. Substituting these facts into $\mathfrak{d}$, we get

$$\mathfrak{d}=-n\left(r+d-\frac{1}{2}rd\right)+\left(d+E-\frac{1}{2}Ed\right). \tag{37}$$

The question is for what value of $\left(r+d-\frac{1}{2}rd\right)$ one expects lower scaling $\mathfrak{d}$ with rising complexity/order of the graph $n$? The answer is for $\left(r+d-\frac{1}{2}rd\right)<0$, so that the first term in Eq. (37) is positive. In this case, larger orders $n$ become less significant (due to the suppression $\alpha^{\mathfrak{d}}$ as $\alpha\to 0$) in the IR, and there is no need to consider complex/large $n$ graphs! We thus conclude that the theory is simplest for

$$r+d-\frac{1}{2}rd\leq 0 \quad\rightsquigarrow\quad d\geq d_c=\frac{2r}{r-2}. \tag{38}$$

For an illustration, let's consider a two-point function of $\phi^4$ theory. In $d=4$, one gets $\mathfrak{d}=2$ (nor relevant in IR) at any order $n$. Thus, the free theory will give the leading infrared behavior.

In $d = 3$, $\mathfrak{d} = -n + 2$; thus, the higher order graphs play an increasingly important role in the IR. The theory is non-classical and would require the calculation of arbitrary complex graphs. Note that the condition (38) coincides with the one from the dimensionality of the coupling constant!

Having learned that, let's return to the discussion of critical points and Yang-Lee edges. We found that near the critical point ($\phi^4$) the theory has an upper critical dimension of four, while near Yang-Lee edge singularity ($\phi^3$), it is 6! This has severe implications, which we will discuss next!

## 3.2 Methods for including fluctuations

Our goal is to locate the Yang-Lee edge singularity in a full theory, including the fluctuations in $d = 3$. What methods are usually applied to study critical phenomena?

- The first method that comes to mind is the $\varepsilon$ expansion. The idea of the method is to consider the system close to but not exactly at the critical dimension. The critical exponents and other universal quantities are independent of $d$ and assume their mean-field values above the upper critical dimension $d_c$. Below $d_c$, universal quantities have a nontrivial dimensionality dependence. Specifically, the critical exponents are smooth, analytical functions of the dimensionality $d$. This observation, made by Wilson and Fisher, led to the formulation of a systematic and controlled RG approach based on an expansion in the small parameter $\varepsilon = d - d_c$. For most of the quantities, $\varepsilon$ expansion is the expansion in the number of loops. For example, for the critical exponent $\beta$ (in Ising universality class) we have[13]

$$\beta = \frac{1}{2} + \underbrace{\frac{1}{6}\varepsilon}_{\text{one loop}} + \underbrace{\frac{1}{162}\varepsilon^2}_{\text{two loop}} + \underbrace{\frac{1}{2}\left(\frac{163}{8748} - \frac{2}{27}\zeta(3)\right)\varepsilon^3}_{\text{three loop}} + \mathcal{O}(\varepsilon^4), \quad \epsilon = 4 - d,$$

where the leading order is the result of the mean-field approximation. By performing the series analysis, one can extrapolate the value of $\beta$ to $d = 3$ ($\varepsilon = 1$). The state of the art, six loop calculations of the critical exponents in $\varepsilon$-expansion can be found in Ref. [18].

The $\varepsilon$-expansion was successful in determining many universal properties. It is tempting to apply this method to computing the universal location of the Yang-Lee edge $z_c$. Here, however, a careful reader might anticipate a problem. As we concluded in the previous subsection, the theory has an upper critical dimension of 6 near Yang-Lee edge, higher than the one of the underlying universality class (4). Thus, it is natural to expect that $\varepsilon$-expansion would fail, as it is designed to work just below the upper critical dimension.[14] This said, nothing prevents one from plowing ahead and computing the location. For the Ising universality class we get (see Ref. [20] for details):

$$|z_c| \approx |z_c^{\text{MF}}| \left[1 + \frac{1}{3}\ln\left(\frac{3}{2}\right)\varepsilon + \underbrace{\varepsilon^2 \log \varepsilon \times (\cdots)}_{\text{all loops contribute}}\right]. \tag{39}$$

---

[13]This can be obtained from $\gamma$ and $\eta$ derived in Ref. [17] and the scaling relation $\beta = \frac{\gamma}{2}\left(\frac{d}{2-\eta} - 1\right)$.

[14]The mean-field approximation of $\phi^4$ theory becomes exact in the limit of $d \to 4$ owing to the diminishing role of the fluctuations – the consequence of vanishing coupling at a fixed point in four dimensions. However, for any value of $t > 0$ and infinitesimally below $d = 4$, fluctuations diverge near the Yang-Lee edge. This divergence arises because the correlation length at the singularity becomes large enough to counteract the smallness of the coupling. See Ref. [19] for more details.

Here we see that the first correction to mean-field is of the order of $\varepsilon$ as expected; however, the next correction contains the logarithm of $\varepsilon$, and most importantly, its coefficient cannot be extracted perturbatively as graphs of all orders contribute to it. Thus, the $\varepsilon$-expansion breaks down as it does not provide a controlled way to extract the location of the Yang-Lee edge.

Note, however, that in contrast to the YLE location, the edge exponent can be and was computed within $\varepsilon$ expansion near 6 dimension (the upper critical dimension of $\phi^3$ theory). For the state of the art, five-loop calculation, see Ref. [21].

- The other widely utilized method is Monte-Carlo simulation on a discretized lattice. Monte-Carlo simulations were very successful in providing first principle results on the critical statics in $d = 3$. However, the calculations at complex values of parameters are impractical due to the sign problem. For illustration, let's consider the Ising model in one dimension, Eq. (21). Monte-Carlo simulations are based on the importance sampling of the probability distribution, for the Ising model, we have

$$P(\sigma) \propto e^{\hat{J} \sum_{i=1}^{L-1} \sigma_i \sigma_{i+1} + \hat{h} \sum_i \sigma_i} . \tag{40}$$

Consider purely imaginary values of $\hat{h} = i\theta$. The probability becomes a complex number. This can be partially fixed by considering combining the probability for $\sigma$ and $-\sigma$ configurations simultaneously:

$$P(\sigma) \propto e^{\hat{J} \sum_{i=1}^{L-1} \sigma_i \sigma_{i+1}} \cos\left( \theta \sum_i \sigma_i \right) . \tag{41}$$

We have an improvement; the probability is no longer a complex number. However, it is not positive semi-definite and thus prevents one from relying on the importance sampling, rendering the approach impractical. This is the essence of the sign problem (see Ref. [22] for an introduction in the QCD context).

Given the failure of these two methods, we have a limited number of options. One of them is the functional renormalization group approach.

## 3.3 Functional renormalization group approach

In the previous subsection, I discussed two conventional methods of studying critical phenomena, and identified two major limitations in locating the Yang-Lee edge singularity. The Functional Renormalization Group (FRG) approach overcomes both limitations and enables one computing the location of Yang-Lee edge directly.

Below, a brief overview of the FRG approach is presented; for a thorough review, see Refs. [23–28]. The FRG is a field-theoretic realization of Wilson's concept of integrating over momentum shells. In FRG, this idea is implemented by sequentially incorporating fluctuations ordered in momentum, starting from the ultraviolet (UV) regime, where the system is governed by a microscopic action, and progressing toward the infrared (IR), which may describe a potentially complex, strongly correlated system. Practically, this is carried out by modifying the path integral with the addition of a mass-like term, $\Delta S_k[\varphi]$, which suppresses contributions from IR momentum modes with $p \lesssim k$. Infinitesimal changes in $k$ yield a differential flow equation that links the UV effective action at a large initial scale $k = \Lambda$, where $\Gamma_{k=\Lambda}[\phi] \approx S[\phi]$, to the full IR effective action at $k = 0$, $\Gamma_{k=0}[\phi] = \Gamma[\phi]$.

Lets consider the partition function in the presence of $\Delta S_k[\phi]$:

$$\mathcal{Z}_k[J] = \int \mathcal{D}\varphi \, e^{-\Delta S[\varphi]} \, e^{-S[\varphi] + J\varphi} . \tag{42}$$

Here, $k$ is a parameter that defines a smooth IR cut off. Most often, one considers a rather obvious choice to suppress IR:

$$\Delta S_k[\varphi] = \frac{1}{2} \int d^d x \int d^d y \; \sum_i \varphi_i(x) R_k(x, y) \varphi_i(y), \tag{43}$$

where $R$ is the regulator function $R_k(x, y) = R_k(x - y)$. Since the goal is to suppress IR modes with $p \ll k$ while keeping UV modes $p \gg k$ unmodified, the regulator function must satisfy the following conditions:

$$R_k(p) \propto k^2, \quad \text{for} \quad p \ll k, \tag{44}$$

$$R_k(p) \to 0, \quad \text{for} \quad p \gg k. \tag{45}$$

A modified Legendre transform leads to the scale-dependent effective action $\Gamma_k[\phi]$

$$\Gamma_k[\phi] = -\ln \mathcal{Z}_k[J] + J\phi - \Delta S_k[\phi]. \tag{46}$$

It is straightforward to show that the functional $\Gamma_k[\phi]$ satisfies the Wetterich equation [23, 29, 30],

$$\partial_k \Gamma_k[\phi] = \frac{1}{2} \text{Tr} \left\{ \partial_k R_k \left( \frac{\delta^2 \Gamma_k[\phi]}{\delta \phi_i \delta \phi_j} + R_k \right)^{-1} \right\}. \tag{47}$$

The initial conditions for $\Gamma_k[\phi]$ are given by the the classical tree-level action in the UV $k = \Lambda$. The solution of this equation at $k = 0$ provides the full quantum action $\Gamma = \Gamma_{k=0}$.

In application to critical statics, the FRG was successfully applied to extract the scaling functions, the critical exponents, and the critical amplitudes for $O(N)$ theories (see, e.g., Refs. [31–48]).

The flow equation represents an infinite hierarchy of coupled partial differential equations for the effective action and its functional derivatives. To solve it practically, a truncation is necessary. The main challenge in FRG is in identifying a small parameter to devise a truncation scheme. Near a critical point, the diverging correlation length enables a systematic expansion around vanishing momentum. It is referred to as the derivative expansion (See Ref. [49] for a formal discussion). Recently, a method to estimate the systematic error of the truncation was developed in Refs. [48, 50]. Due to the numeric complexity, only a modest next-to-leading order of this expansion, i.e., first order in momentum-squared, was applied to locate the Yang-Lee edge.[15]

To briefly clarify this numerical challenge: numerical implementation of the FRG entails solving a set of integro-differential equations. For critical exponents, it is only necessary to identify a fixed point of these equations, which simplifies the problem to a set of algebraic equations involving integrals. In contrast, locating the Yang-Lee edge requires solving the full set of integro-differential equations. To make this computation manageable on current hardware, Refs. [20, 51] employed a specific form of the FRG cut-off function (the Litim regulator), allowing for the analytical evaluation of integrals and reducing the problem to the partial differential equations. The flip side of this regulator choice is that it limits the derivative expansion to no higher than next-to-leading order. Combined with computational constraints, this restriction did not let us reach the same precision in the YLE calculations as has been attained for critical exponents in Refs. [48, 50].

---

[15]The critical exponents were computed to next-to-next-to-leading order of the derivative expansion in Ref. [48].

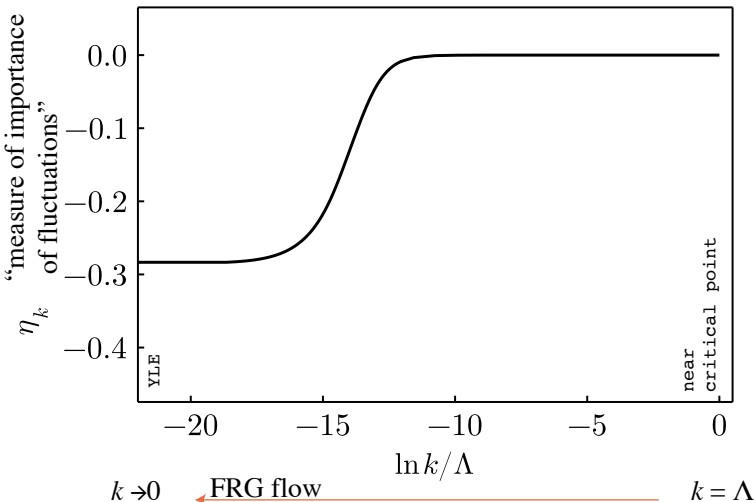

Figure 7: The effective anomalous dimension for Ising universality class and $d = 4$ as a function of the cut-off momentum $k$. The flow equation tracks the position of the Yang-Lee edge singularity at $m_R = 0+$. In the infrared limit ($k \to 0$), the anomalous dimension approaches its physical value at the Yang-Lee edge singularity. The magnitude of $\eta_k$ can be interpreted as the measure of the importance of fluctuations; at $d = 4$, near the Wilson-Fisher point, $\eta = 0$ indicates that fluctuations can be neglected (the mean-field approximation is exact for describing long-range phenomena). As the system nears the Yang-Lee edge fixed point, the absolute value of the anomalous dimension increases significantly, demonstrating that the upper critical dimension of the Yang-Lee edge fixed point (6) is higher than that of the Wilson-Fisher fixed point (4). See Ref. [51] for the details of the calculation.

## 3.4 Yang-Lee edge with FRG: Collection of results

Here I briefly summarize the results of Refs. [20, 51], where the authors considered $O(N)$ universality classes for all $N$ of practical interest ($N = 1$ corresponds to the Ising universality class).

At the next-to-leading order of the derivative expansion, the truncation of the effective action reads

$$\Gamma_k[\phi] = \int d^d x \left( U_k(\phi) + \frac{1}{2} Z_k(\phi)(\partial_i \phi)^2 \right).$$

The flow equations can be reformulated as a set of equations for the effective potential $U$ and the wave function renormalizations $Z$. For the average potential, one gets

$$\partial_t U_k(\rho) = \frac{1}{2} \int \bar{d}^d q \, \partial_t R_k(q^2) \left[ G_k^{\parallel} + (N-1) G_k^{\perp} \right], \quad \rho = \frac{\phi^2}{2},$$

with

$$G_k^{\perp} = \frac{1}{Z_k^{\perp}(\rho)q^2 + U_k'(\rho) + R_k(q^2)}, \quad G_k^{\parallel} = \frac{1}{Z_k^{\parallel}(\rho)q^2 + U_k'(\rho) + 2\rho U_k''(\rho) + R_k(q^2)}.$$

Here we introduced the longitudinal and the transverse (Goldstone) modes.

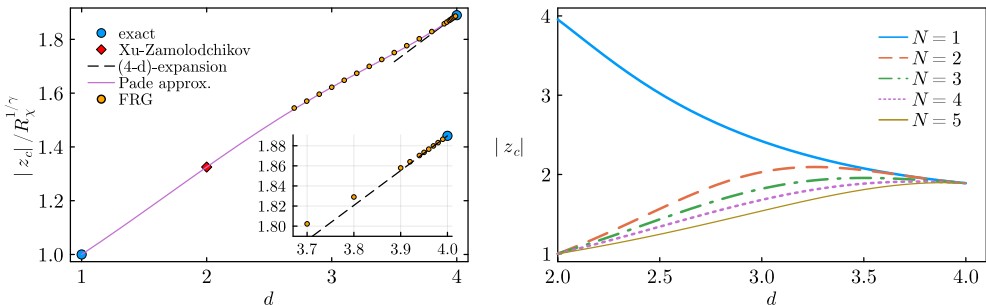

Figure 8: Left panel: the universal location of the Yang-Lee edge singularity as a function of the number of dimensions $d$ for N=1. The location of the Yang-Lee edge singularity in the two-dimensional Ising model (diamond) is obtained from the result of Ref. [52]. The locations in one- and four-dimensions are known analytically. See Ref. [51] for details. Right panel: four-parameter Padé approximation for the dependence of the Yang-Lee edge location on the number of spatial dimensions for different numbers of field components, $N$; see Ref. [20] for details.

The flow equation for the wave function renormalization of the longitudinal mode reads

$$
\partial_t Z_\parallel(\phi) = \int \bar{d}^d q \, \partial_t R_k(q^2) \bigg\{ G_\parallel^2 \Big[ \gamma_\parallel^2 \big(G_\parallel' + 2G_\parallel'' \frac{q^2}{d}\big) + 2\gamma_\parallel Z_\parallel'(\phi)\big(G_\parallel + 2G_\parallel' \frac{q^2}{d}\big)
$$
$$
+ (Z_\parallel'(\phi))^2 G_\parallel \frac{q^2}{d} - \frac{1}{2} Z_\parallel''(\phi) \Big]
$$
$$
+ (N-1) G_\perp^2 \Big[ \gamma_\perp^2 \big(G_\perp' + 2G_\perp'' \frac{q^2}{d}\big) + 4\gamma_\perp Z_\perp'(\phi) G_\perp' \frac{q^2}{d} + (Z_\perp'(\phi))^2 G_\perp \frac{q^2}{d}
$$
$$
+ 2\frac{Z_\parallel(\phi) - Z_\perp(\phi)}{\phi} \gamma_\perp G_\perp - \frac{1}{2}\Big(\frac{1}{\phi} Z_\parallel'(\phi) - \frac{2}{\phi^2}(Z_\parallel - Z_\perp)\Big) \Big] \bigg\}.
$$

A similar equation for the transverse mode can be found in Ref. [20]. In the above set of equations we used the shorthand notations for

$$
\gamma_\parallel = q^2 Z_\parallel'(\phi) + U^{(3)}(\phi), \quad \gamma_\perp = q^2 Z_\perp'(\phi) + \frac{\partial}{\partial \phi}\Big(\frac{1}{\phi} U'(\phi)\Big), \quad G' = \frac{\partial G}{\partial q^2}.
$$

In summary, the flow equation leads to three coupled partial-differential equations on $U$, $Z_\parallel$ and $Z_\perp$. The equations are stiff and solving them is rather challenging. Instead, one considers Taylor series expansion near $k$-dependent expansion point $\phi_k$ and corresponding equations for the expansion coefficients and $\phi_k$. Traditionally the expansion point is selected to coincide with the minimum of the effective potential $U_k'[\phi_k] = h = \text{const}$. To locate the Yang-Lee edge, this is however is not the best choice – it is better to expand near $\phi_k$ defined by $U_k''[\phi_k] = m^2 \to 0$. Then, the FRG equation tracks the critical manifold in the parameter space and thus interpolates between the Wilson-Fisher fixed point and the Yang-Lee edge singularity as illustrated in Fig. 7. In the figure, I show the evolution of the effective anomalous dimensions.[16] Without going into details, one can consider the effective anomalous dimension to be a measure of the importance of fluctuations. When fluctuations are not important $\eta = 0$. The dashed line ($d = 4$) in this figure demonstrates the fact we considered before: near the critical fixed point, $\eta = 0$, but as we approach Yang-Lee edge fixed point the magnitude of $\eta$ increases and reaches a quite large value $|\eta| \approx 0.3$. This is a manifestation of the fact that

---

[16]The effective anomalous dimension assumes the value of the actual anomalous dimension at fixed points.

Table 1: The location of the Yang-Lee edge singularity, $|z_c|/R_\chi^{1/\gamma}$ and $|z_c|$ for a representative number of components $N$ in three dimensions. For $|z_c|/R_\chi^{1/\gamma}$, the numbers in the parentheses show the truncation error and the error due to residual regulator dependence. For $|z_c|$, the number in the parentheses represents a combined uncertainty, including that originating from the universal amplitude ratio $R_\chi$ and the critical exponent $\gamma$. $R_\chi$ and $\gamma$ are taken from Refs. [31,54] and Refs. [48,55]. See Ref. [20] for more details.

| $N$ | 1 | 2 | 3 | 4 | 5 |
|---|---|---|---|---|---|
| $|z_c|/R_\chi^{1/\gamma}$ | 1.621(4)(1) | 1.612(9)(0) | 1.604(7)(0) | 1.597(3)(0) | 1.5925(2)(1) |
| $|z_c|$ | 2.43(4) | 2.04(8) | 1.83(6) | 1.69(3) | 1.55(4) |

the upper critical dimension of the Yang-Lee edge is larger than the one of the conventional critical point.

The results are presented in Table 1. Note that our approach gives direct access to the universal ratio $|z_c|/R_\chi^{1/\gamma}$ not $|z_c|$. To find $|z_c|$, we used known results for the universal amplitude ratio $R_\chi$ and the critical exponent $\gamma$. Remarkably, FRG calculations can be performed for a non-integer number of dimensions, see Fig. 8. The left panel of the figure demonstrates that $|z_c|/R_\chi^{1/\gamma}$ is consistent with results obtained at the linear order of $\varepsilon$-expansion, Eq. (39) and with two- and one-dimensional Ising models. The right panel of Fig. 8 shows $z_c$ dependence on the number of dimensions and the number of the field components $N$. The infinite $N$ limit for the location of the singularity is known exactly.[17] Fig. 8 shows that the approach to this limit is logarithmically slow. Reference [53] demonstrated that $|z_c|$ depends on $N$ non-monotonically and the asymptotic behavior is set in for $N \gg 10$.

We established the essential part of the analytical structure near a phase transition, but to have a complete picture, we also need to investigate the symmetries of the QCD partition function. We turn to this next.

# 4 QCD analytic structure

The transition from hadronic gas to quark-gluon plasma along the finite $T$ axis and zero chemical potential was shown to be a smooth "crossover". Model calculations, functional methods, and recent analysis of lattice results (described below) hint on the existence of the critical point at some nonzero value of the baryon chemical potential. The crossover changes to a first-order phase transition at the critical point. There is no rigorously defined order parameter to describe the transition. However, due to the smallest of the light quark masses, the chiral condensate is the closest proxy for the order parameter. From the previous discussion, it should be clear that the critical point comes with the corresponding Yang-Lee edge singularities. To distinguish them from other Yang-Lee edges potentially present in QCD, I will refer to them as critical end point Yang-Lee edge singularities.

## 4.1 Roberge-Weiss periodicity

In the introduction, we discussed the properties of the free fermion distribution function including periodicity and singularities in the complex plane of the chemical potential. It is necessary to generalize these results to QCD, where the situation is similar qualitatively but quan-

---

[17]The magnetic equation of state in the infinite $N$ limit is given by $f_G(f_G^2 + z)^2 = 1$. An interested reader can easily locate the singularity by finding the solution of the magnetic equation of state and $dz/df_G = 0$.

titatively different. To proceed, consider the QCD partition function at the purely imaginary chemical potential

$$Z(T, \mu = i\theta T) = \text{Tr} e^{-\frac{\hat{H}}{T} + i\theta \hat{N}}. \tag{48}$$

Due to particle-antiparticle symmetry $Z(T, -\theta) = Z(T, \theta)$. The partition function can be rewritten as a functional integral [6]:

$$Z(T, \theta) = \int Dq D\bar{q} DA e^{-S_E(T, \theta)}, \tag{49}$$

$$S_E = \int_0^{1/T} d\tau \int d^3x \left( \bar{q}(\gamma^\mu D_\mu - m)q - \frac{1}{4}G^2 - i\theta T q^\dagger q \right). \tag{50}$$

The boundary conditions for the fermionic fields are anti-periodic in the Euclidean time direction $q(x, \tau = 1/T) = -q(x, 0)$ and periodic for the gluon field.

The explicit dependence of the Euclidean action on the imaginary chemical potential can be removed by performing the change of variables $q(x, \tau) \to e^{i\theta T\tau}q(x, \tau)$. Although the boundary conditions are not invariant under the variable change

$$q(x, \tau = 1/T) = -e^{i\theta}q(x, \tau = 0), \tag{51}$$

this form of the partition function enables us to learn more about QCD symmetries.

Additionally, consider an aperiodic gauge transformation

$$q(x, \tau) \to U(x, \tau)q(x, \tau), \tag{52}$$

$$A(x, \tau) \to U(x, \tau)A(x, \tau)U^{-1}(x, \tau) - \frac{i}{g}\partial_\mu U(x, \tau)U^{-1}(x, \tau). \tag{53}$$

Let's choose $U$ such as to satisfy the boundary condition

$$U(x, \tau = 1/T) = e^{2\pi i k/N_c}U(x, \tau = 0), k = 1, \ldots, N_c. \tag{54}$$

This special class of aperiodic gauge transform is symmetric up to elements of $Z(N_c)$.

As with any gauge transformation, the action is invariant under Eq.(54). Moreover, the boundary conditions for the gluon field are invariant under the transformation due to a general property: the adjoint representation is invariant under the center of the group ($Z(N_c)$ in this case). In contrast to the gluon field, the quark field boundary conditions change:

$$q(x, \tau = 1/T) = -e^{i\left(\theta + \frac{2\pi}{N_c}k\right)}q(x, \tau = 0). \tag{55}$$

Comparing Eqs. (51) and (55), we conclude that

$$Z(T, \theta) = Z\left(T, \theta + \frac{2\pi}{N_c}k\right). \tag{56}$$

This is the so-called Roberge-Weiss symmetry. For three colors, we thus have the symmetry $\theta \to \theta + 2\pi/3$ or more generally $\hat{\mu} \to \hat{\mu} + 2\pi i/3$. Note that this is different from the symmetry of the free fermion partition function: $\hat{\mu} \to \hat{\mu} + 2\pi i$.

Similar to free fermions, we expect to have a singularity (we will refer to it as the Roberge-Weiss Yang-Lee edge) somewhere on the line $\hat{\mu} = \hat{\mu}_r + \frac{i\pi}{3}$, as illustrated in Fig. 9.

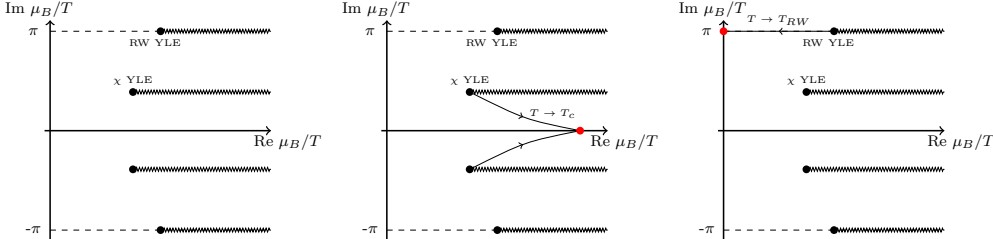

Figure 9: An illustration of the complex plane of the baryon chemical potential and the expected location of the Yang-Lee edges associated with the critical endpoint and Roberge-Weiss transition. The panel on the left corresponds to the temperature in the range $T_c < T < T_{RW}$. When the temperature is lowered $T \to T_c$, the critical endpoint Yang-Lee edge singularities approach and eventually pinch the real axis at $T = T_c$, as demonstrated by the middle panel. The right panel illustrates the case when temperature is increased $T \to T_{RW}$. Again, two corresponding Yang-Lee edge singularities (note that the figure is symmetric under $\mu_B \to -\mu_B$ approach and pinch imaginary axis.)

## 4.2 Complex chemical potential plane

Now we are ready to assemble a complete albeit schematic picture of the complex baryon chemical potential plane (see also Ref. [14]). For temperatures in the range $T \in [T_c, T_{RW}]$, we expect to have

- The Roberge-Weiss Yang-Lee edge singularities on the lines $\mathrm{Im}\,\hat{\mu}_B = 3\,\mathrm{Im}\hat{\mu} = \pm\pi$ ($\pm$ is due to the Roberge-Weiss periodicity).

- The Yang-Lee edge singularities associated with the critical endpoint are located in the complex plane.

We also need to remember that generically the complex plane has the reflection symmetry about both $\mathrm{Im}\hat{\mu}_B$ and $\mathrm{Re}\hat{\mu}_B$ axes. What happens to the position of the critical end point Yang-Lee edge when the temperature is lowered towards $T_c$? The singularity approaches the real chemical potential axis, and at $T_c$, two critical end point Yang-Lee edge singularities pinch the real axis. This is illustrated by the middle panel in Fig. 9. Now consider the Roberge-Weiss Yang-Lee edge. As the temperature is increased towards $T_{RW}$, the singularity slides along the line $\mathrm{Im}\,\hat{\mu}_B = \pm\pi$ and at $T_{RW}$, two Roberge-Weiss Yang-Lee edge singularities pinch the imaginary chemical potential axis (note that $\mu_B = 3\mu$, as $\mu$ denotes the quark chemical potential). This is illustrated in the right panel of Fig. 9 (only one singularity is shown in the figure).

Lattice QCD calculations indicate that at $T = T_{RW}$, the Roberge-Weiss critical point coincides with critical point probed by the chiral condensate. This means that the critical endpoint Yang-Lee edge either approaches the Roberge-Weiss singularity as $T \to T_{RW}$ or joins it at some temperature lower than $T_{RW}$ (neither scenario is shown in the figure). It is currently unknown, and there is no compelling argument supporting either option.

## 5 Review of the results based on lattice calculations

Lattice QCD simulations do not allow us to locate the Yang-Lee edge singularities directly due to the sign problem. However, one can estimate their location using analytic continuation (e.g., via Padé approximation). Assuming that the location of the Yang-Lee edge is found,

next, we can analyze its trajectory as a function of temperature and potentially establish the existence/location of the corresponding critical point. As far as the critical endpoint Yang-Lee edge is concerned, an additional complication arises from the lack of high-precision lattice data at low temperatures, where we expect the critical point to reside. Thus, one must use the Yang-Lee edge scaling based on Eq. (14) and extrapolate the Yang-Lee edge trajectory towards the real baryon chemical potential. Below, we detailed this strategy. However, it is important to validate the strategy for the case when the location of the Yang-Lee edge is known. This is why I start with the Roberge-Weiss Yang-Lee edge.

## 5.1 Roberge-Weiss Yang-Lee edge

The key question is if, by tracking the corresponding YLE singularities, one can locate the position of the critical point in actual lattice QCD calculations. In application to Roberge-Weiss critical point, this question is of no direct interest, as one can find the critical temperature by analyzing lattice data at imaginary baryon chemical potential.[18] However, testing the methods to locate and track the Roberge-Weiss singularity constitutes a robust validation of the approach.

At $T < T_{\rm RW}$, the Roberge-Weiss Yang-Lee edge is located at chemical potential with both non-zero real and imaginary parts. Thus direct determination of the location of the singularity in QCD is not possible. The authors of Ref. [58, 59] perform the calculations at purely imaginary values of the baryon chemical potential and then analytically continue the results into the complex plane using the Padé approximation for the first and second derivatives of pressure ($p/T^4$) with respect to the chemical potential ($\hat{\mu}_B$). For the first derivative, the Padé approximant was taken in the following form

$$\chi_1^B(\hat{\mu}_B) = \frac{\sum_{i=0}^m a_i \hat{\mu}_B^i}{1 + \sum_{j=1}^n b_j \hat{\mu}_B^j} \,. \tag{57}$$

Since the left-hand side of this equation can be computed on the lattice for different imaginary values of $\hat{\mu}_B$, determining the coefficients $a_i, b_i$ simply requires solving a set of linear differential equations for each temperature value. Once the coefficients are known, the zeroes of the denominator can be used to determine the closest singularity $\hat{\mu}_{\rm YLE}$. It approximates the position of the Yang-Lee edge in the infinite volume limit.

The next step is to analyze the trajectory of this singularity as a function of temperature. Equation (14), Yang-Lee scaling, provides a framework for this. To make use of it, $t$ and $h$ have to be expressed in terms of the QCD thermodynamic parameters $T$ and $\mu_B$. In the vicinity of the Roberge-Weiss critical point (for which the critical chemical potential is known $\hat{\mu}_B = i\pi$), we have the following mapping relations

$$\frac{t}{t_0} = \frac{T_{\rm RW} - T}{T_{\rm RW}} \,, \qquad \frac{h}{h_0} = \frac{\hat{\mu}_B - i\pi}{i\pi} \,, \tag{58}$$

where $t_0, h_0$ are non-universal metric constants and $T_{\rm RW}$ is the Roberge-Weiss phase transition temperature. From Eq. (14), we thus have

$$\hat{\mu}_{\rm RW\ YLE}(T) = a \left( \frac{T_{\rm RW} - T}{T_{\rm RW}} \right)^{\beta\delta} \,, \tag{59}$$

where the parameter $a$ is a combination of $t_0$, $h_0$ and $|z_c|$. It is to be determined by the fit alongside $T_{\rm RW}$. There is one subtle point to be taken into account before one determines $T_{\rm RW}$:

---

[18]Note that there is no sign problem in QCD at purely imaginary values of the baryon chemical potential, see Refs. [56, 57].

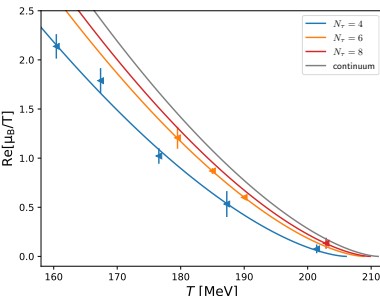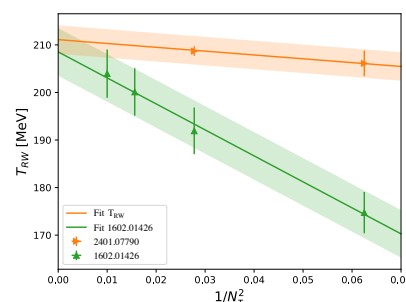

Figure 10: Left panel: scaling of the Yang-Lee singularity with temperature for three different lattice spacings ($N_\tau$) and continuum extrapolation, see Refs. [58, 59] for details. Right panel: the $N_\tau$ dependence of the Roberge-Weiss transition temperature $T_{\text{RW}}(N_\tau)$ from Refs. [58, 59] and direct calculation of Ref. [60].

lattice cut-off dependence or the dependence on the $N_\tau$. By fitting parameters, $a$ and $T_{\text{RW}}$ in equation (59) one finds them at a given value of $N_\tau$. In Ref. [59], it was assumed that the cut-off dependence of the Roberge-Weiss critical temperature and the amplitude is given by

$$T_{\text{RW}}(N_\tau) = T_{\text{RW}}^{(0)} + T_{\text{RW}}^{(2)}/N_\tau^2 \,, \tag{60}$$

$$a(N_\tau) = a^{(0)} + a^{(2)}/N_\tau^2 \,, \tag{61}$$

which is justified for large enough $N_\tau$. All together, one ends up with four fitting parameters. The left panel of Fig. 10 demonstrates the quality of the scaling (59) at a given value of $N_\tau$, while the right panel shows the extrapolation of $T_{\text{RW}}$ to the continuum limit, $T_{\text{RW}}^{(0)} = 211.1 \pm 3.1$ MeV. Comparison with the direct calculations at the imaginary value of the chemical potential from Ref. [60] shows a good agreement with the result obtained by tracing the Yang-Lee edge, see the right panel of Fig. 10.

This serves as a proof of principle for locating the critical point by tracing the location of the Yang-Lee edge. Can a similar strategy be executed for a QCD critical point at a finite real chemical potential? The answer is yes, as the recent articles [61, 62] demonstrated. I will now summarize their methods and results.

## 5.2 Yang-Lee edge associated with critical end point

As I alluded to before, the main challenge of locating the QCD critical point with the current lattice data can be attributed to the lack of the low-uncertainty low-temperature calculations. For example, the current usable data at imaginary chemical potential reaches $T \approx 120$ MeV. At the same time, there is a strong indication from the functional methods that the critical point is at about 100 MeV. Thus, the Yang-Lee edge trajectory cannot be found near the possible critical point, and one has to rely on the extrapolation of the Yang-Lee edge trajectory to lower temperatures. Together with the statistical uncertainty of the lattice data and the systematic error of the analytical continuation, the extrapolations constitute the major sources of uncertainty for determining the location of the critical point.

I describe the extrapolation of the trajectory first, as it is common for both strategies described below. It is similar in idea to the analysis of the Roberge-Weiss singularity but different in the actual realization. Assume we could locate the Yang-Lee edge at a set of temperatures $T > T_c$. This input allows us to fix unknown non-universal numbers in the relation (14). For this, we need to establish the relationship between $t$ and $h$, the actual temperature, and the chemical potential. In the case of the critical point, we have a mixture of the reduced temperature $\Delta T = T - T_c$ and the reduced chemical potential $\Delta\mu = \mu - \mu_c$ contributing to the scaling

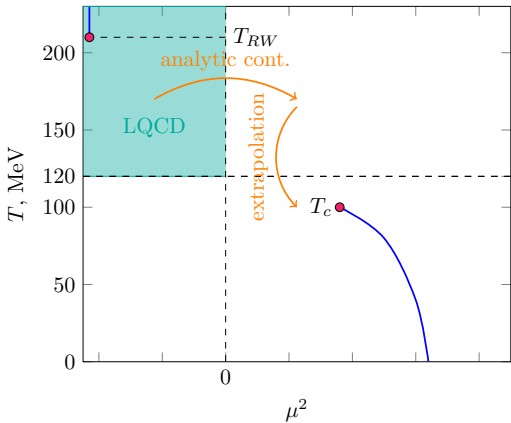

Figure 11: The figure illustrates the domain of temperature and chemical potential where Lattice QCD performed direct simulations: imaginary (negative $\mu^2$) and zero baryon chemical potential and at temperatures approximately above 120 MeV. To access thermodynamics at real values of the chemical potential (positive values of $\mu^2$), an analytic continuation has to be performed. It has to be supplemented by some physics-motivated extrapolation technique to access temperatures lower than 120 MeV.

variables

$$h = a\Delta T + b\Delta\mu, \tag{62}$$

$$t = c\Delta T + d\Delta\mu. \tag{63}$$

Substituting to Eq. (14) we have

$$a\Delta T + b\Delta\mu = i|z_c|^{\beta\delta}(c\Delta T + d\Delta\mu)^{\beta\delta}. \tag{64}$$

Considering small $\Delta T$ and $\Delta\mu$ and taking into account that for Z(2) universality class $\beta\delta \approx 3/2 > 1$ at the leading order, we have

$$a\Delta T + b\Delta\mu = 0,$$

or

$$\mathrm{Re}\,\Delta\mu_{\mathrm{LO}} = -\frac{a}{b}\Delta T. \tag{65}$$

Thus, the real part of the reduced chemical potential approaches zero linearly as a function of temperature. The imaginary part of the reduced chemical potential can be easily computed by substituting Eq. (65) to (64) and taking the imaginary part. The leading imaginary part is then

$$\mathrm{Im}\,\Delta\mu_{\mathrm{LO}} = \frac{1}{b}|z_c|^{\beta\delta}\left(c - \frac{da}{b}\right)^{\beta\delta}\Delta T^{\beta\delta}. \tag{66}$$

That is, the imaginary part of the reduced chemical potential approaches zero faster than the real part. Of course, there will be higher-order corrections for both equations. The data analysis suggests that one has to add $\Delta T^2$ to the real part to fit $\mathrm{Re}\,\Delta\mu$ as a function of temperature. For the sake of simplicity, it is convenient to lump together different combinations of constants into new coefficients in front of different powers of $\Delta T$:

$$\mathrm{Re}\,\mu = \mu_c + c_1\Delta T + c_2\Delta T^2, \tag{67}$$

$$\mathrm{Im}\,\mu = c_3\Delta T^{\beta\delta}. \tag{68}$$

All in all, there are five unknown coefficients $c_1$, $c_2$, $c_3$, $T_c$ and $\mu_c$. They have to be fixed using the lattice data.

SciPost Phys. Lect. Notes 91 (2025)

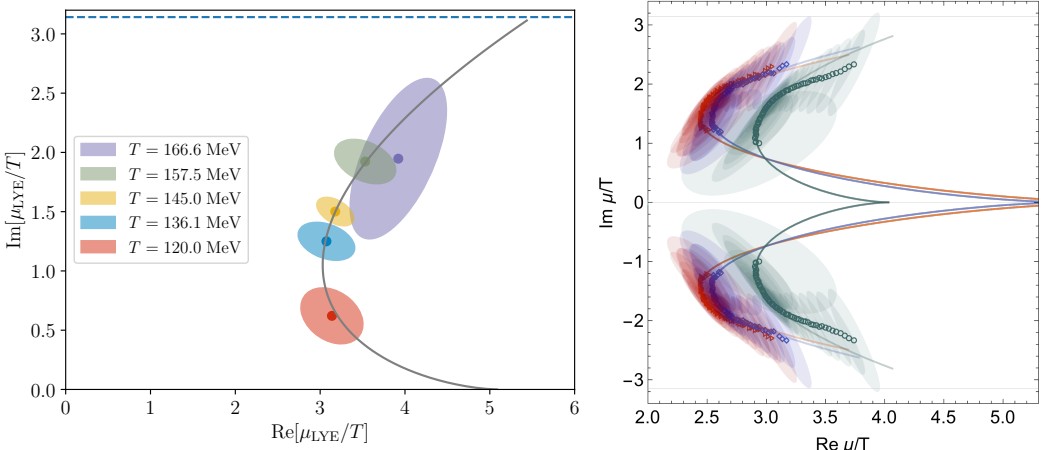

Figure 12: Left panel: the figure is from Ref. [62]. It shows the location of the Yang-Lee edge singularity as a function of temperature for $N_\tau = 6$ extracted using the imaginary baryon chemical potential data. The solid line is the result of the best fit with Eq. (68). Right panel: the figure is from Ref. [61]. In the figure, the location of the Yang-Lee edge singularity is displayed as a function of temperature extracted from Taylor series coefficients for $N_\tau = 8$. The Yang-Lee edge locations (data points) and corresponding fits (solid lines) correspond to different extraction methods. The red triangles and the blue diamonds are obtained by utilizing two different conformal maps with subsequent Padé approximation. The green circles are obtained by using Padé without any conformal map.

### 5.2.1 Purely imaginary chemical potential data

The general strategy is the same as the one described in Sec. 5.1. That is, the lattice data is collected at imaginary values of the baryon chemical potential and then analytically continued into the complex plane using multi-point Padé. The closest zero of the denominator in Padé approximant estimates the position of the Yang-Lee edge. Of course, the data must be collected at lower temperatures, and the fit must be performed with Eq. (68). The fit of the imaginary $\mu_B$ data was done within the interval $[\hat{\mu}_{B\min}, \hat{\mu}_{B\max}] \in [-i\pi, i\pi]$. The length of the interval $|\hat{\mu}_{B\min} - \hat{\mu}_{B\max}|$ was varied between $\pi$ and $2\pi$. Overall, the authors of Ref. [62] perform 55 Padé approximations per one temperature value. Only zeroes of the denominator closest to $[\hat{\mu}_{B\min}, \hat{\mu}_{B\max}]$ were used. This provides a handle on the systematic uncertainty of the extraction of the Yang-Lee edge singularity. I refer the reader to Ref. [62] for more technical details. The results are presented in Fig. 12. The data was analyzed for $N_\tau = 6$ and 8.[19] The largest $N_\tau$ yields the location of the critical point at $T_c = 101 \pm 15$ MeV, $\mu_c = 560 \pm 140$ MeV. As the authors point out, the continuum limit will most likely result in larger chemical potential values of about 650 MeV [62] and slightly higher temperatures of about 110 MeV [63].

### 5.2.2 Taylor series data

Analysis of Ref. [61] is similar, but there are the following key differences

- Taylor series expansion coefficients computed at zero chemical potential were used. Note that currently, only eight coefficients are known. The highest two of them lack the continuum limit estimate and are computed at a fixed $N_\tau = 8$.

---

[19]$N_\tau = 8$ data was analyzed using the single point Padé, see Ref. [62] for details.

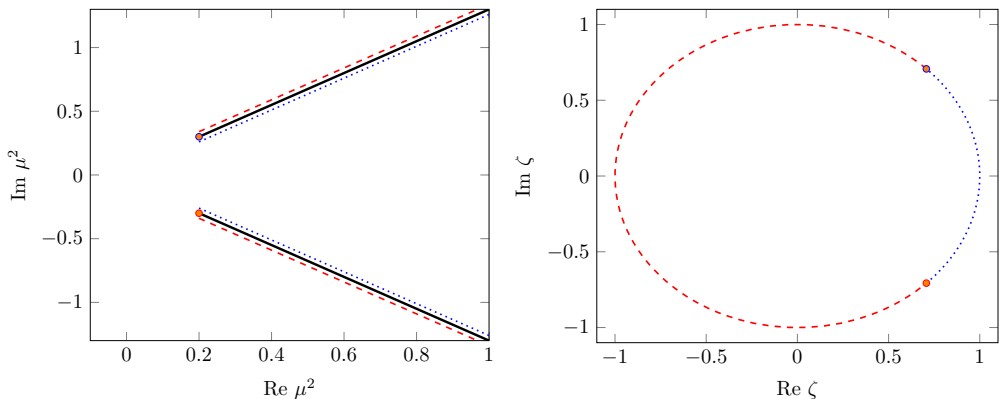

Figure 13: Illustration of the "two-cut" map of Ref. [61].

- Application of the Padé analysis is done after the conformal mapping applied to help improve the extraction of singularity. That is, first, one performs the change of the variable $\mu^2 = f(\zeta)$, next obtains the Taylor series in $\zeta$ about $\zeta_0$ defined by $f(\zeta_0) = 0$, and finally performs Padé in $\zeta$ variable. The reason to perform a conformal map before Padé analysis is that when the function is expected to have a complex conjugate pair of singularities, Padé results in two arcs of poles that emerge from singularities; they coalesce along the real axis. These poles limit the radius of convergence of the Padé approximant. The idea is to map them beyond the disk of the radius defined by the Yang-Lee edge.

The function $f$ is chosen such as to map the YLE cuts onto the unit circle. A figure best demonstrates this; see Fig. 13. The red and blue lines are mapped onto the unit circle. The function that performs this "two-cut" transformation is

$$f(\zeta) = 4\left|\mu_{\text{YLE}}^2\right| \zeta \left[\frac{\theta/\pi}{(1-\zeta)^2}\right]^{\theta/\pi} \left[\frac{1-\theta/\pi}{(1+\zeta)^2}\right]^{1-\theta/\pi}. \tag{69}$$

This transformation has to be supplied by $\mu_{\text{YLE}} = |\mu_{\text{YLE}}|e^{i\theta}$. This is, however, exactly what we want to find. The author of Ref. [61] adopted the following strategy: first, a rough estimate for the Yang-Lee edge location is found using the result of pure Padé analysis, then it is improved by applying the map, expanding in $\zeta$ and performing the Padé analysis of the corresponding Taylor series in $\zeta$. This procedure is then iterated until it converges to a fixed point that defines $\mu_{BYLE}$. Beyond this, the uniformizing map was performed in Ref. [61]. However, the result is very close to the "two-cut" map and thus will not be described here.

In Fig. 12, the result of this procedure is shown. Blue diamonds show the two-cut map, while the pure Padé gives green circles.

**An example of improving Taylor series convergence with a conformal map**

It seems that the first step (conformal map) plays a crucial role, and to demonstrate one of the advantages of performing a conformal map for the convergence of the Taylor series, let us return to the example we considered in Introduction, Eq. (4). We will perform the following set of conformal maps. Firstly, we map $\mu$ to the fugacity plane

$$\lambda = e^{\hat{\mu}}.$$

This map accounts for the periodicity of our function and transforms the lines $\hat{\mu} = \pm i\pi$ to the negative real axis. Secondly, we map the negative real axis onto the circumference of the unit

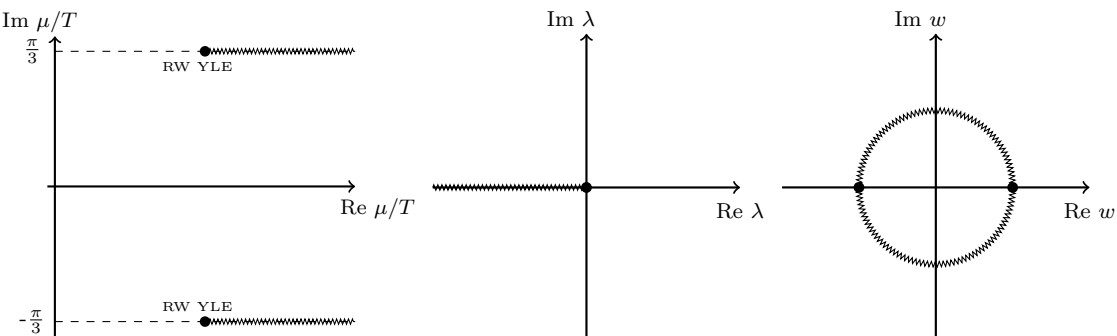

Figure 14: Series of maps Eq. (70) and Eq. (71). As a result, the strip $-\pi < \mathrm{Im}\,\hat{\mu} < \pi$ is mapped onto the unit disk $|w| < 1$.

circle using

$$w = \frac{\sqrt{\lambda} - 1}{\sqrt{\lambda} + 1}, \tag{70}$$

with the inverse

$$\lambda = \frac{(w+1)^2}{(w-1)^2}. \tag{71}$$

The expansion about $\hat{\mu} = 0$ corresponds to the expansion about $w = 0$. For our example, we set $\hat{\omega} = 1$ and obtained the radius of convergence in $\hat{\mu}$ to be $R_{\hat{\mu}} = \sqrt{1 + \pi^2}$. In the variable $w$, the radius of convergence is $R_w = 1$. That is, after mapping, the function can be extracted to values of $\hat{\mu}$:

$$\left| \frac{e^{\hat{\mu}/2} - 1}{e^{\hat{\mu}/2} + 1} \right| < 1,$$

or simply

$$\left| e^{\hat{\mu}/2} - 1 \right| < \left| e^{\hat{\mu}/2} + 1 \right|,$$

which is true for any $\hat{\mu}$, thus allowing us to recover our function from Taylor series expansion for any $\hat{\mu}$. By doing the maps (see illustration in Fig. 14), we gained a lot: it became possible for us to extend the finite range of convergence to an unbounded one. We achieved this by incorporating information about the analytic structure of our function into the map.

To demonstrate this explicitly, start with a truncated series expansion in $\hat{\mu}$ ($a_i$ is non-zero for even $i$ only):

$$f(\hat{\mu}) = \sum_{i=0}^{M} a_i \mu^i,$$

Next, substitute $\mu = \ln \frac{(w+1)^2}{(w-1)^2}$ (the inverse of the two maps performed) into this expression and expand in $w$ about zero (due to $w(\mu = 0) = 0$) to the same truncation order:

$$f(w) = \sum_{i=0}^{M} b_i w^i,$$

where $w$ is a function of $\mu$, $w = \frac{e^{\hat{\mu}/2} - 1}{e^{\hat{\mu}/2} + 1}$. Finally, plot the resulting $f(\mu)$. This procedure is illustrated in Fig. 15 for $M = 20$. We see a significant improvement in reproducing the exact

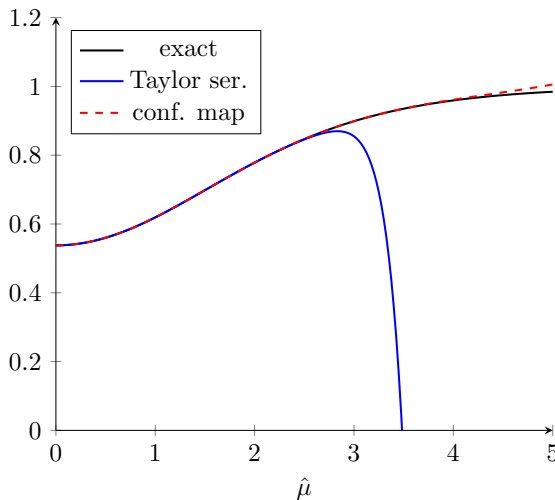

Figure 15: Illustration of the convergence of the truncated Taylor series expansion in $\bar{\mu}$ and the one in $w$ after performing the conformal map. Both series are truncated at order 20. The figure demonstrates how a conformal map can improve the convergence of the Taylor series without providing any extra Taylor series coefficients.

function. Of course, there is a nuance. We assumed that the coefficients $a_i$ are known precisely and that linear combination does not lead to an exploding uncertainty. To study this question, let's consider a few terms in the expansion of $f(w)$ after performing the conformal map and reexpanding the series; we have

$$f(w) = a_0 + 16a_2 w^2 + \left(\frac{32}{3}a_2 + 256a_4\right)w^4 + \dots$$

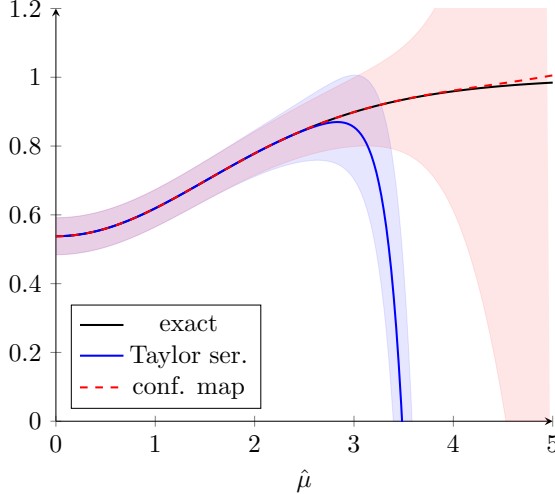

Figure 16: The same as in Fig. 15, but with 10% uncertainty added to the original Taylor series coefficients.

This demonstrates that indeed the coefficients $b_j$ are just linear combinations of $a_{i \leq j}$, specifically

$$
\begin{aligned}
b_0 &= a_0, \\
b_2 &= 16a_2, \\
b_4 &= \frac{32}{3} + 256a_4, \\
&\cdots
\end{aligned}
$$

If $a_i$ are known with some finite precision, one might get concerned about large coefficients appearing in $b_2$ in front of $a_2$ or $b_4$ in front of $a_2$ and $a_4$. However, in this specific case, this would be unfounded. Consider the quadratic term, $a_2 \hat{\mu}^2$ for $\hat{\mu} = 5$. It equals $25a_2$. At the same time, taking into account that $w(\hat{\mu} = 5) \approx 0.85$ we have $16a_2 w^2 \approx 12a_2$. Thus, for the quadratic truncation, the actual coefficients in front of $a_2$ are not larger for the series expansion after the map. However, at higher orders, we have more terms contributing to a $b_j$, and the situation with the uncertainty estimate can drastically change. This is more straightforward to demonstrate with an actual computation. I will continue working with the same function but prescribe a 10% uncertainty to each coefficient $a_i \to a_i \pm a_i/10$. I also assume there is no correlation between the uncertainties for each coefficient. After this, the analysis can be repeated. Its result is shown in Fig. 16. The figure demonstrates that most of our gains with the conformal map are wiped away by a 10% uncertainty in determining the coefficients $a_i$. This example should be taken as a caution tale; it cannot provide a universal conclusion of the usefulness of a conformal map for the case when Taylor series coefficients are known with a given finite precision.

# 6 What else Yang-Lee edge singularity affects?

## 6.1 Asymptotic Taylor series

In the previous section, I reviewed that the location of the Yang-Lee edge singularity can be estimated by analyzing Taylor series coefficients. One can go a step further in this direction analytically and demonstrate that the location and the edge critical exponent are imprinted in high order Taylor series coefficients due to Darboux theorem [64, 65]. To demonstrate this, consider a function $f(x)$ with singularity closest to the origin located at the real point $x = x_1$. Suppose that the behavior of this function near the singularity[20] is

$$
f(x) = (1 - x/x_1)^{\sigma+1} r(x) + a(x), \tag{72}
$$

where $r(x)$ is an analytic in some disk centered at $x_1$ and having radius larger than $|x_1|$.

$$
r(x) = \sum_{k=0}^{\infty} b_k (x - x_1)^k, \tag{73}
$$

Similarly, $a(x)$ is assumed to be analytic in some disk with center $x = 0$ and radius $> |x_1|$. The critical exponent $\sigma$ is non-integer (the case of interest for us). The assumed form of the function in Eq. (72) is quite restrictive but is sufficient for our purposes.

Expanding $(1 - x/x_1)^{k+\sigma+1}$ into a power series about $x = 0$, we have

$$
(1 - x/x_1)^{k+\sigma+1} = \sum_{n=0}^{\infty} \frac{\Gamma(n-k-1-\sigma)}{\Gamma(n+1)\Gamma(-1-k-\sigma)} \left(\frac{x}{x_1}\right)^n.
$$

---

[20]This form neglects the confluent singularity.

Combining this equation with Eq. (73), we get

$$(1-x/x_1)^{\sigma+1} r(x) = \sum_{n=0}^{\infty} \sum_{k=0}^{\infty} (-1)^k b_k x_1^{k-n} \frac{\Gamma(n-k-1-\sigma)}{\Gamma(n+1)\Gamma(-1-k-\sigma)} x^n .$$

That is, the asymptotic coefficients of the expansion are given by

$$c_n = \sum_{k=0}^{\infty} (-1)^k b_k x_1^{k-n} \frac{\Gamma(n-k-1-\sigma)}{\Gamma(n+1)\Gamma(-1-k-\sigma)} . \tag{74}$$

In general, the expansion coefficients of the function $f(x)$ are the sum of $c_n$ and $a_n$ ($a(x) = \sum a_n x^n$). However, the asymptotic expansion coefficients of $f(x)$ are not affected by $a_n$ and are given by $c_n$ only. To show this, consider the closest singularity of $a(x)$ to the expansion point $x = 0$ at some $x = x_2$ with $|x_2| > |x_1|$. Compared to $c_n$, $a_n$ are smaller by a factor $O(|x_1/x_2|^n)$, and hence are negligibly small asymptotically.

Applying Stirling's formula for $n \to \infty$ to the ratio

$$\frac{\Gamma(n-k-1-\sigma)}{\Gamma(n+1)} = n^{-k-\sigma-2} \left( 1 + \frac{(k+\sigma+1)(k+\sigma+2)}{2n} + O\left(\left(\frac{1}{n}\right)^2\right) \right) ,$$

demonstrates that successive terms in Eq. (74) are suppressed by inverse powers of $n$. For sufficiently large $n$ we thus have

$$c_n \sim b_0 x_1^{-n} \frac{\Gamma(n-1-\sigma)}{\Gamma(n+1)\Gamma(-1-\sigma)} \sim \frac{b_0 x_1^{-n}}{n^{2+\sigma}\Gamma(-1-\sigma)} . \tag{75}$$

In QCD, as we discussed above, there are two complex conjugate singularities in the $\hat{\mu}_B^2$ plane. Both are the same distance from the expansion point (the origin). Therefore, the above analysis has to be amended. This can be easily done in the same fashion as above with the result:

$$\begin{aligned} c_n &\sim \frac{b_0 x_1^{-n}}{n^{2+\sigma}\Gamma(-1-\sigma)} + \text{c.c.} \\ &= \frac{2|b_0||x_1|^{-n}}{n^{2+\sigma}\Gamma(-1-\sigma)} \cos(\phi_{b_0} - n\phi_{x_1}) , \end{aligned} \tag{76}$$

where $\phi_{b_0}$ and $\phi_{x_1}$ are the phases of $b_0$ and $x_1$ respectively. In application to QCD we interpret $x \to \hat{\mu}_B^2$ and $x_1 \to \hat{\mu}_{\text{BYLE}}^2$.

We thus conclude that the location of the Yang-Lee edge and its critical exponent $\sigma$ define the asymptotic behavior of Taylor series coefficients. Of course, to gain anything practical from this analysis in QCD, one has to compute high-order Taylor series expansion coefficients. As I pointed out before, the highest available Taylor series coefficients from lattice QCD calculations are of a modest eighth order. At what order the asymptotics of Eq. (76) set in is unknown and would require detailed analysis, most likely involving higher order Taylor coefficients.

## 6.2 Fourier coefficients

Another striking example of the Yang-Lee edge influence on thermodynamics is the Fourier coefficient of the baryon chemical potential. As we established in Sec. 5.1, the QCD partition function is periodic in baryon chemical potential: $Z(\hat{\mu}_B + 2\pi i) = Z(\hat{\mu}_B)$. Due to this periodicity, analysis of the data obtained for purely imaginary values of baryon chemical potential is natural in terms of the corresponding Fourier coefficients [67–72]. Usually one computes Fourier transformation of the baryon number density $n_B = V^{-1}\partial_{\hat{\mu}_B} \ln Z(\hat{\mu}_B)$:

$$b_k = \frac{1}{i\pi} \int_{-\pi}^{\pi} d\theta \, \hat{n}_B(\hat{\mu}_B = i\theta), \sin(k\theta) , \tag{77}$$

where I explicitly took into account the symmetry property of the baryon density $n_B(\hat{\mu}_B) = -n_B(-\hat{\mu}_B)$ and introduced $\hat{n}_B = n_B/T^3$.

The Fourier coefficients are sensitive to the structure of the phase diagram through the following line of arguments, see Ref. [66] for details. Consider the integral (77) in the complex plane, see Fig. 17 for an illustration. Since the continuous deformation of the contour does not change the value of the integral as long as it does not cross the singularities, one can modify the integration to that over both sides of the cuts, as shown in Fig. 17. To proceed with an actual realization of this program, it is helpful first to consider just one singularity and the corresponding cut in isolation.

Consider an odd function $n_B(\hat{\mu}_B)$ periodic in an imaginary argument having brunch points in the complex plane located at $\pm\hat{\mu}_B^{\rm br}$, where $\hat{\mu}_B^{\rm br} = \hat{\mu}_r^{\rm br} + i\,\hat{\mu}_i^{\rm br}$. Here, we expand $n_B$ near the branch point $\hat{\mu}_B \to +\hat{\mu}_B^{\rm br}$:

$$\hat{n}_B(\hat{\mu}) = A(\hat{\mu}_B - \hat{\mu}_B^{\rm br})^\sigma(1 + B(\hat{\mu}_B - \hat{\mu}_B^{\rm br})^{\theta_c} + \ldots) + \sum_{n=0}^{\infty} a_n(\hat{\mu}_B - \hat{\mu}_B^{\rm br})^n, \tag{78}$$

with $\sigma > -1$ and $\theta_c > 0$. In the context of the YLE singularity, $\theta_c$ is the confluent critical exponent (not to be confused with integration variable $\theta$). The regular part of $\hat{n}_B$ on the cuts is encoded by the coefficients $a_n$.

From the definition of the Fourier coefficients, we have

$$b_k = \frac{1}{\pi}\int_{-\pi}^{\pi} d\theta\, \hat{n}_B(\hat{\mu}_B = i\theta).e^{-ik\theta}. \tag{79}$$

To compute the integral, we will deform the contour as shown in Fig. 18. The figure assumes that the right-most points are extended to infinity, i.e., $\mathrm{Re}\,\mu \to \infty$. The contribution of the segments $(ab)$ and $(gh)$ cancel each other due to the integrand's periodicity and the segments' opposite direction. The contributions from $(bc)$ and $(fg)$ is zero due to the exponential decay

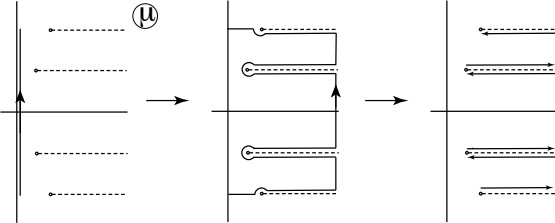

Figure 17: Complex chemical potential plane with the critical end point and RW YLE. The integration path along the imaginary chemical potential axis can be deformed so that the contour follows around the branch point singularities and the cuts. The figure is from Ref. [66].

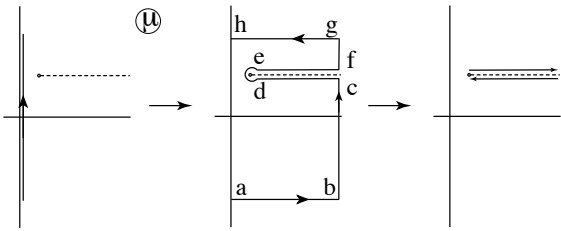

Figure 18: Computation of the Fourier coefficient for the function with one singularity in the right half-plane of complex baryon chemical potential. The figure is from Ref. [66].

of $\exp(-ik\theta) = \exp(-k\hat{\mu}_B)$ for any $k > 0$. The integral around the branch point $(de)$ vanishes due to $\sigma > -1$. Thus, only the segments on both sides of the cut $(cd)$ and $(ef)$ give a non-trivial contribution to the integral, as illustrated in Fig. 18. To evaluate the contribution of these segments, we consider the parametrization $i\theta = s + \hat{\mu}_B^{\text{br}}$

$$\frac{1}{\pi} \int_{(ef)} d\theta \, \hat{n}_B(\hat{\mu}_B = i\theta) e^{-ik\theta} = \frac{1}{i\pi} e^{-\mu_B^{\text{br}}k} \int_0^\infty ds \, \hat{n}_B(\hat{\mu}_B = s + \hat{\mu}_B^{\text{br}}) e^{-ks} . \tag{80}$$

Evaluating the integrals, one obtains

$$\frac{1}{\pi} \int_{(ef)} d\theta \, \hat{n}_B(\hat{\mu}_B = i\theta) e^{-ik\theta} = \frac{e^{-\mu_B^{\text{br}}k}}{i\pi} \left( A \frac{\Gamma(1+\sigma)}{k^{1+\sigma}} \left[ 1 + \frac{B}{k^{\theta_c}} \frac{\Gamma(1+\sigma+\theta_c)}{\Gamma(1+\sigma)} + \dots \right] + \sum_{n=0}^\infty a_n \frac{\Gamma(1+n)}{k^{1+n}} \right).$$

In the second line, the sum is due to the analytic part in Eq. (78).

The integral over the segment $(cd)$ is identical to the expression above except for the $2\pi$ rotation around the branch point and an extra minus sign due to the direction of the segment. Adding both integrals together cancels the analytic part to yield

$$b_k = \frac{e^{-\mu_B^{\text{br}}k}}{i\pi} A \frac{\Gamma(1+\sigma)}{k^{1+\sigma}} \left( 1 - e^{i2\pi\sigma} + \frac{B}{k^{\theta_c}} \left[ 1 - e^{i2\pi(\sigma+\theta_c)} \right] \frac{\Gamma(1+\sigma+\theta_c)}{\Gamma(1+\sigma)} + \dots \right). \tag{81}$$

Absorbing $k$ independent factors into constants $A$ and $B$, we arrive at

$$b_k = \tilde{A} \frac{e^{-\mu_B^{\text{br}}k}}{k^{1+\sigma}} \left( 1 + \frac{\tilde{B}}{k^{\theta_c}} + \dots \right). \tag{82}$$

Now, we are ready to generalize this result to the case when both YLE and RW singularities are present, we obtain

$$b_k = \tilde{A}_{\text{YLE}} \frac{e^{-\hat{\mu}_B^{\text{YLE}}k}}{k^{1+\sigma}} \left( 1 + \frac{\tilde{B}_{\text{YLE}}}{k^{\theta_c}} + \dots \right) + \tilde{A}_{\text{RW}} \frac{e^{-\hat{\mu}_B^{\text{RW}}k}}{k^{1+\sigma}} \left( 1 + \frac{\tilde{B}_{\text{RW}}}{k^{\theta_c}} + \dots \right) + \text{c.c.} \tag{83}$$

The coefficients $\tilde{A}_{\text{YLE,RW}}$ and $\tilde{B}_{\text{YLE,RW}}$ are generally complex numbers. Taking into account that $\text{Im} \, \hat{\mu}_B^{\text{RW}} = \pi$, we have

$$b_k = |\tilde{A}_{\text{YLE}}| \frac{e^{-\hat{\mu}_r^{\text{YLE}}k}}{k^{1+\sigma}} \left( \cos(\hat{\mu}_i^{\text{YLE}}k + \phi_a^{\text{YLE}}) + \frac{|\tilde{B}_{\text{YLE}}|}{k^{\theta_c}} \cos(\hat{\mu}_i^{\text{YLE}}k + \phi_b^{\text{YLE}}) + \dots \right)$$

$$+ |\hat{A}_{\text{RW}}|(-1)^k \frac{e^{-\hat{\mu}_r^{\text{RW}}k}}{k^{1+\sigma}} \left( 1 + \frac{|\hat{B}_{\text{RW}}|}{k^{\theta_c}} + \dots \right), \tag{84}$$

where $\phi_a$ and $\phi_b$ are phases due to non-trivial phases of $\tilde{A}^{\text{YLE}}$ and $\tilde{B}^{\text{YLE}}$ and trivial real factors were absorbed into $|\hat{A}_{\text{RW}}|$ and $|\hat{B}_{\text{RW}}|$.

A few comments about the obtained results:

- The coefficients, $b_k$, are exponentially sensitive to the imaginary values of the positions of the Yang-Lee edge singularities and sensitive to the edge critical exponent $\sigma$.

- The confluent critical exponent $\theta_c = \nu_c \omega = \frac{\sigma+1}{3}\omega$ is about 0.6 ($\omega$ can be found in Ref. [21]) and thus leads to an appreciable suppression of the corresponding terms. It is safe to drop these corrections for a large enough order of $k$.

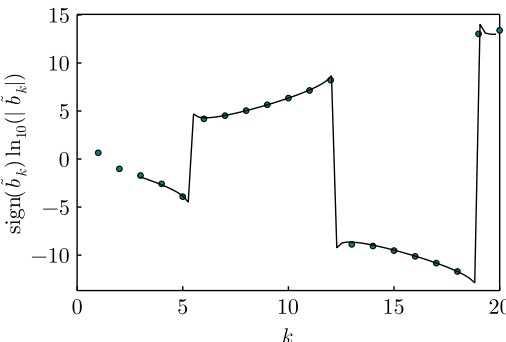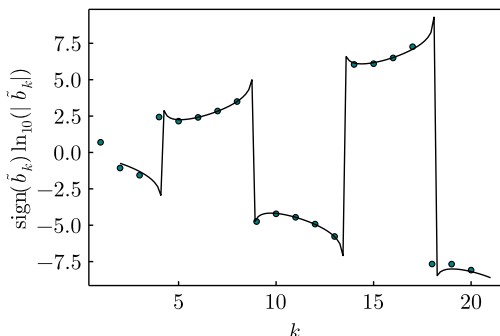

Figure 19: LPA FRG Fourier coefficients $\tilde{b}_k = k^{1+\sigma} b_k$ ($\sigma_{\text{LPA}} = 1/5$) and the corresponding fits for $T = 150$ MeV (left) and $T = 180$ MeV (right). The figure is from Ref. [66].

- Consider a free fermion of mass $m$ and spin $1/2$. To the arbitrary order in the degeneracy (see e.g. Ref. [6]) we have

$$\hat{n}_b = \frac{2}{\pi^2} \hat{m}^2 \sum_k \frac{(-1)^{k+1}}{k} K_2(k\hat{m}) \sin(k\hat{\mu}). \tag{85}$$

The Fourier coefficients of this expansion is trivial to obtain. Using the asymptotic expansion of the modified Bessel function, $K_2(x) \approx \left(\frac{\pi}{2x}\right)^{1/2} e^{-x}$, one finds

$$b_k \propto \frac{1}{k^{3/2}} e^{-k\hat{m}}. \tag{86}$$

That coincides with the leading order of the last term of Eq. (84) with the expected $\sigma = 1/2$ and $\hat{\mu}_r^{\text{RW}} = \hat{m}$.

Again, as for the Taylor coefficients, in QCD, it is not known at what order $k$ the asymptotic behavior sets in. A model calculation can explore this. Here, I consider the quark-meson model computed using FRG for the local potential approximation. This approximation leads to $\sigma_{\text{LPA}} = 1/5$ (about a factor of two larger than expected, but a significant improvement from the mean-field value of $\sigma_{\text{MF}} = 1/2$). The model is described in detail in Ref. [73]. For more information on the calculations, see Ref. [66].

Figure 19 shows the dependence of the Fourier coefficients on the order $k$ for two temperatures. The asymptotic form Eq. (84) fits this dependence and extracts the location of the Yang-Lee edge. As one can see from the figure, the asymptotic behavior sets in for rather modest values of $k$. The fit yields $\hat{\mu}_{\text{YLE}}^{\text{fit}} = 1.483(7) + i\,0.446(6)$ ($\hat{\mu}_{\text{YLE}}^{\text{fit}} = 0.949(8) + i\,0.675(11)$) for $T = 150(180)$ MeV. The actual location of the singularity can be found using FRG, and it is $\hat{\mu}_{\text{YLE}} = 1.553 + i\,0.4794$ ($\hat{\mu}_{\text{YLE}} = 0.9445 + i\,0.6618$) for $T = 150(180)$ MeV. Although it was not possible to reliably extract the Fourier coefficient for $k > 22$, the fit accuracy is sufficiently high as it reproduces the location within 5% precision.

The model results do not capture the first principle lattice QCD calculations, where the asymptotics might set in at larger values of $k$. Most importantly, model results provide high-precision data for $n_B(\mu_B)$, while lattice QCD results have unavoidable statistical uncertainty, which may significantly affect the extraction of $b_k$.

# 7 Summary

In these lecture notes, I provided an introduction to the analytic structure of the equation of state near a second-order phase transition with a specific focus on the Yang-Lee edge singularity. I reviewed the results of recent papers on the universal location of the Yang-Lee edge for different universality classes. In relation to QCD, I tried to show why understanding the analytic structure of QCD may be a key to locating the critical point and establishing the equation of state for non-zero values of the baryon chemical potential. I reviewed very recent results on the location of the critical point obtained through the analysis of the location of the Yang-Lee edge extracted by the analytic continuation of the imaginary baryon chemical potential data and Taylor series coefficients computed on the lattice. Finally, I also discussed how the location of the Yang-Lee edge and its properties manifest themselves in the asymptotics of baryon number Fourier coefficients and Taylor series coefficients.

# Acknowledgments

I am grateful to S. Tiwari for carefully reading these notes. I thank the organizers of the 2024 XQCD Ph.D. school and the conference for their support and specifically Yi Yin for hospitality.

**Funding information** This work is supported by the U.S. Department of Energy, Office of Nuclear Physics through contract DE-SC0020081.

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
