# Peer review of "Two lectures on Yang-Lee edge singularity and analytic structure of QCD equation of state"

_SciPost Physics Lecture Notes, doi:SciPost Phys. Lect. Notes 91 (2025)_

## Round 1 · Referee Report · Anonymous (Referee 1) · 2025-2-5

Report

The manuscript, written for a course held at the 2024 XQCD PhD school that took place shortly before the conference of the same name, discusses the analytic structure of the equation of state near a second-order transition and the Yang-Lee edge singularities (YLEs), with a focus on the analytic structure of QCD.

The topic is highly relevant to current research, as demonstrated by the recent publication of several papers on the subject (some of which are reviewed in the manuscript) and numerous presentations at international conferences.

The lecture notes are well written. The presentation is systematic and clear. YLEs are discussed within and beyond the mean-field approximation. Special attention is given to practical applications. The author reviews recent developments in the field, such as the determination of the Roberge-Weiss Yang-Lee edge singularities, the tracking of YLEs trajectories to pinpoint the location of the critical endpoint of QCD, an improvement of the analysis using conformal maps. The author also discusses how the YLEs affect the Taylor series expansion coefficients and the Fourier coefficients of the baryon number density.

I think the manuscript meets the journal's acceptance criteria and can be published after a minor revision. These lecture notes will be valuable to future graduate and PhD students.

Requested changes

Minor remarks:

1- I believe in eq. 5 there is an extra minus in the second passage (and a missing minus in the first passage):

   $f(\hat{\mu}) = \frac{ 1}{e^{   \hat{\omega} - \hat{\mu}_r + i \pi} +1} =  -\frac{ 1}{1-e^{   \hat{\omega} - \hat{\mu}_r }} = - f_B(\hat{\mu}_r) $

$\to$ $f(\hat{\mu}) = \frac{ 1}{e^{ \hat{\omega} - \hat{\mu}_r - i \pi} +1} = \frac{ 1}{1-e^{ \hat{\omega} - \hat{\mu}_r }} = - f_B(\hat{\mu}_r) $

Moreover, at the beginning of pg. 4:

"[$\hat{\mu}^* = \hat{\omega} \pm i\pi n$] where $n$ is a positive integer"

$\to$ "[$\hat{\mu}^* = \hat{\omega} \pm i\pi n$] where $n$ is a positive odd integer"

2- Section 5.2.1 outlines the multi-point Pade' analysis conducted in Ref 55 and explains how the systematic uncertainty stemming from the interval dependence was evaluated by building Pade' approximations for 55 distinct intervals. After describing this strategy, the author writes that the data was analyzed for $N_t=6,8$ and quotes the $N_t=8$ estimate for the location of the CEP. However, in Ref 55 the multi-point Pade' analysis was applied to $N_t=6$ only. The $N_t=8$ estimate was determined using a single-point Pade' approximation, a similar but different strategy, with no interval dependence. Maybe it could be made more clear that the $N_t=6$ estimate was determined using the strategy outlined in the section, whereas the $N_t=8$ estimate was determined using a different strategy (yielding a result compatible within errors with the $N_t=6$ estimate, but with slightly larger $\mu$ and $T$ likely due to cut-off effects).

3- I spotted some typos:

  • at the end of section 1, ": This" $\to$ ": this"

  • at the beginning of section 2.3,"mean-filed" $\to$ "mean-field"

  • at the end of pg. 18, "Fif. 8" $\to$ "Fig. 8"

  • at the end of pg. 21,"simply require" $\to$ "simply requires"

  • at pg. 26, "$R_\omega$" $\to$ "$R_w$"

  • at pg. 27, "$\omega(\mu=0) = 0$" $\to$ "$w(\mu = 0) = 0$" "$b_i\omega^i$" $\to$ "$b_i w^i$" "$\omega$ is a function of $\mu$, $\omega =$" $\to$ "$w$ is a function of $\mu$, $w =$"

  • at the beginning of section 6.2, "of the baryon chemical potential" $\to$ "of the baryon number density"

  • sometimes the imaginary part of a complex number has an extra $i$ in front (axes in Figs. 9,14 and equations in section 4.2)

Recommendation

Ask for minor revision

---

## Round 1 · Referee Report · Anonymous (Referee 2) · 2025-2-11

Report

I have reviewed mainly sections 2 and 3 of the lecture notes. Below I give suggestions for improvements.

In general, I found the exposition sometimes vague and lacking references. While it's acceptable to omit derivations in lecture notes, references should be complete to provide students with entry points to the literature.

Examples where references are lacking: - Section 2.3, p.7: "one can prove that the scaling variable is given..." - Section 2.4, p.9: "M. Fischer dubbed Yang-Lee edge singularities protocritical points" - p.10: "Every finite temperature critical point has the associated pair of Yang-Lee edges"

Here are a few places where the discussion was unclear to me:

Section 2.5: "The location of the singularity is universal." In what sense is it universal? Let us take two different models in the Ising model universality class, e.g., the nearest-neighbor Ising model and some of its next-to-nearest interaction extensions. Will the location of the singularity be the same in both models? Or is the universality meant in some other sense?

This central point has not been explained. Also, it would be helpful to explain clearly why computing z is so important.

Section 3.2 p.14: The equation for beta in the ε-expansion is an expansion of a critical exponent, while the expansion of (39) is for z_c, which is not a critical exponent. Thus, one is comparing apples and oranges regarding the goodness of these expansions.

Critical exponents for the Ising universality class have been computed in the epsilon expansion up to six or seven loops (Kompaniets and Panzer; Schnetz - it would be good to provide a reference to the state of the art) and, after appropriate Borel resummation, this gives results in good agreement with Monte Carlo simulations. For LY, I am aware of an old paper by O. F. de Alcantara Bonfim, J.E. Kirkham and A.J. McKane, "Critical exponents to order ε³ for φ³ models of critical phenomena in 6−ε dimensions", J.Phys. A 13 (1980) L247, but I don't know if this was updated and what is the status of Borel resummation and comparison to Monte Carlo simulations (comparing e.g., to the lattice animals https://arxiv.org/pdf/cond-mat/0408061).

The issue seems to be not so much that epsilon must be small for the epsilon expansion to work, as this range can be expanded with appropriate resummations, but that in the expansion for z_c, one encounters non-analytic behavior and infinitely many diagrams enter beyond some order in epsilon. Can one provide a physical explanation for why this is the case?

Recommendation

Ask for minor revision

---

## Round 1 · Referee Report · Anonymous (Referee 3) · 2025-2-13

Report

These lecture notes are an introduction to the concept of Lee-Yang edge singularities and their use in the context of determining the critical point of QCD at finite temperature and chemical potential.

The notes are pedagogical and well written. I recommend their publication. I have few short remarks, that the author may take into account to improve the document.

Requested changes

Below eq (20), I don't understand why we need to adjust 4 parameters to reach a tri-critical point, and not 3.

Below eq (39) it would be very valuable to explain in few sentences why the epsilon expansion fails for computing the location of the edge singularity. The author may reassure the reader by stating that the computation of critical exponents is safe in this model and maybe quote the 3-loop expression for eta or sigma.

In section 3.4, it is not clear at the beginning why the calculation
is performed at a generic N, and not N=1 because the emphasis was on
Ising univarsality class in the preceding
sections.

page 17, the anomalous dimension is clearly negative in the plot but
is given to be +0.3 in the main text.

page 18, table 1 at would be valuable to give the values of
gamma. Also, the table should mention the dimension relevant to these
results.

I think the author should clarify at the beginning of section 4 that
there is no sign problem in QCD when the chemical potential is purely
imaginary, and maybe cite one of the publications of de Forcrand and
Philipsen on this topic.

Typos:
Above Eq. (32), the bar on hbar is not placed correctly
Just before section 4, fif->fig
above eq (57), Padè->Padé

Recommendation

Publish (meets expectations and criteria for this Journal)

---

## Round 2 · Author Response

To referees: I thank you for your feedback, constructive comments, and careful reading of my manuscript. Your insights were extremely helpful, and I have applied the feedback where appropriate in the manuscript. I believe I addressed all the issues raised by all three reviewers, with just two exceptions.

Referee 1 wrote ``at the beginning of section 6.2, "of the baryon chemical potential" → "of the baryon number density"``. In this sentence I indeed meant baryon chemical potential.
Referee 3 suggested adding the $\gamma$ critical exponent to Table 1. I considered this but decided against it for consistency, as I would also need to include $R_\chi$. Adding two additional rows might distract the reader, given that these quantities are not the main focus of this lecture note. Instead, I have included references to the papers from which the data for $R_\chi$ and $\gamma$ were taken.

---

## Round 2 · List of Changes

Figures: Fig 9 and Fig 14 was modified as suggested be Referee 1: the extra "i" was removed from the y axes.

Equations:
1) To make equations a bit simple and more transparent, I rewrote Eq. (39) for Ising universality class only (that is substituted $N=1$).
2) I corrected Eq. (5), as the first referee suggested.

Here are some minor text changes (this list does not include trivial typos): Page 4: In the beginning of the page "positive integer" -> "positive odd integer, as suggested by Referee 1. Page 7: Reference [7] was added in a paragraph before Eq. (15), as suggested by Referee 2. Page 8: A paragraph (at the very top of the page) on the universality of the magnetic equation of state and the location of the YLE singularity was added, as requested by Referee 2. Page 9: In the middle of the page, Reference [12] was added to the sentence "For this reason, M. Fisher dubbed Yang-Lee edge singularities protocritical points [12]", as was requested by Referee 2. Page 10: Before equation 25. Reference [10] was added, as was requested yb Referee 2. Page 14: At the end of the first paragraph the sentence " The state of the art, six loop calculations of the critical exponents in ϵ-expansion can be found in Ref. [18]." was added as requested by the Referee 2 . Page 14: A discussion on the calculation of the edge exponent in epsilon expansion was added one paragraph below Eq.(39) as requested by the Referee 2: "Note, however, that in contrast to the YLE location, the edge exponent can be and was computed within ϵ expansion near 6 dimension (the upper critical dimension of φ^3 theory). For the state of the art, five-loop calculation, see Ref. [21]." Page 14: As was requested by Referees 2 and 3, footnote 14 was added to address why epsilon expansion fails for the location of the YLE. Reference [19] was added as well.
Page 18: The following text was added to the table caption: Rχ and γ are taken from Refs. [31, 54] and Refs. [48, 55]. Page 20: Extra factors of "i" were removed when referring to "Im" parts of various quantities, as Referee 1 suggested. Page 24: Footnote 19 was added to clarify that $N_\tau=8$ data was computed using single point Pade, as Referee 3 suggested. Page 27: $\omega$ was switched to $w$ everywhere where appropriate, as Referee 1 suggested.

---

## Editorial Decision

published